# CAI: Caption-Sensitive Attention Intervention for Mitigating Object Hallucination in Large Vision-Language Models

## Abstract

Although Large Vision-Language Models (LVLMs) have demonstrated remarkable performance on downstream tasks, they frequently produce contents that deviate from visual information, leading to object hallucination. To tackle this, recent works mostly depend on expensive manual annotations and training cost, or decoding strategies which significantly increase inference time. In this work, we observe that LVLMs' attention to visual information is significantly enhanced when answering caption queries compared to non-caption queries. Inspired by this phenomenon, we propose **C**aption-sensitive **A**ttention **I**ntervention (**CAI**), a training-free, plug-and-play hallucination mitigation method that leverages the attention activation pattern corresponding to caption queries to enhance LVLMs' visual perception capability. Specifically, we use probing techniques to identify attention heads that are highly sensitive to caption queries and accurately estimate optimized intervention directions for their outputs. This intervention strengthens LVLM's fine-grained visual perception capabilities, thereby effectively mitigating object hallucination. CAI reduced object hallucination by an average of 6.03% across five widely used LVLMs and five benchmarks including both discriminative and generative tasks, demonstrating state-of-the-art (SOTA) performance while incurring little additional inference cost and preserving other foundational capabilities.

## 1 Introduction

Despite the remarkable performance of Large Vision-Language Models (LVLMs) on downstream tasks, it is widely observed that LVLMs frequently generate content that conflicts with the corresponding visual information, leading to object hallucination (Sahoo et al., 2024; Huang et al., 2023). To tackle this, recent works for mitigating hallucination mostly use contrastive decoding strategies (Leng et al., 2024; Zhong et al., 2024) which arise high inference latencies, or training LVLMs using carefully designed data (You et al., 2023; Yu et al., 2024a) which incurs expensive manual annotation and computation cost. Furthermore, interpretability studies (Arif et al., 2025; Bi et al., 2024a) have identified insufficient attention to visual information as an underlying cause of hallucination. To address the aforementioned limitations and the underlying cause, we focus on exploring how to enhance LVLMs' perception capability by providing sufficient attention to visual information, without modifying model parameters or introducing significant inference cost.

In this work, we observe that caption query (e.g."Please describe this image in detail.") is a special type of instruction that plays a critical role in LVLM's pre-training stage for text-image alignment, endowing the model with fine-grained visual perception capability. Furthermore, as shown in Figure 1 (a) and (b), we reveal a critical phenomenon: visual attention across particular attention heads is significantly enhanced when fed caption queries versus non-caption queries. We term these attention heads as *caption-sensitive attention heads*. As an enhancement of their visual attention is accompanied by a reduction in object hallucination, it may indicate that these heads are responsible for the fine-grained perception capabilities. Inspired by this phenomenon, we propose **C**aption-sensitive **A**ttention **I**ntervention (**CAI**), a training-free, plug-and-play method, which probes and refines caption-sensitive attention heads outputs during inference to enhance LVLM's fine-grained visual perception capability and mitigate object hallucination. Specifically, our method unfolds in

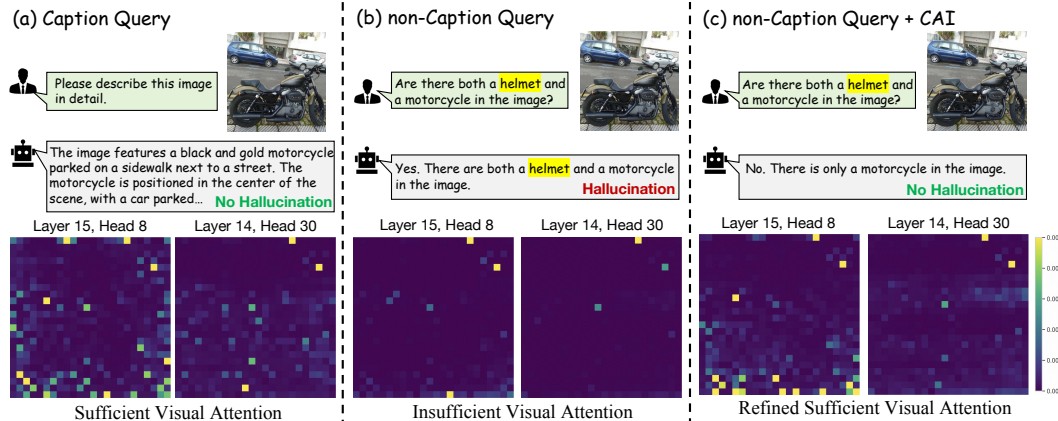

Figure 1: The visualization of attention weights at image patch level across different conversation settings. LLaVA-1.5-7b correctly generates the detailed content of the image in response to the caption query, but exhibits hallucination (e.g., "helmet") when answering the non-caption query. CAI refines LVLM's visual attention patterns from insufficient to sufficient, effectively enhancing visual perception capability and mitigating object hallucination.

three steps. First, following prior work (Li et al., 2024), we use probing techniques to identify these caption-sensitive attention heads. Furthermore, we compute attention output shift vectors for these attention heads, which quantify the output differences from non-caption to caption queries and serve as a fine-grained perception optimization direction. Finally, we apply the precomputed shift vectors to intervene caption-sensitive attention heads during inference, steering their outputs toward a state optimized for fine-grained visual perception and effectively mitigating object hallucinations. As shown in Figure 1 (b) and (c), CAI leads to a notable enhancement in visual attention and effectively mitigates object hallucination.

Consistent improvement across five widely used LVLMs and five benchmarks demonstrates that CAI achieves state-of-the-art (SOTA) performance. On the POPE (Li et al., 2023b) benchmark, the accuracy and the F1 score improve by 5.14% and 5.50% on average. Furthermore, hallucination rates decrease by 7.8% on the MMHalBench (Sun et al., 2023), while the informativeness of the responses improves.

In summary, our main contributions are three-fold:

- Our work is the first to explicitly reveal the impact of caption queries versus non-caption queries on the attention activation patterns of LVLMs, providing novel insights for the optimization of visual attention.

- We propose **CAI**, a training-free method that effectively mitigates object hallucination by refining caption-sensitive attention head outputs during inference with little additional inference cost.

- Comprehensive experimental results demonstrate that CAI not only mitigates hallucination effectively but also shows strong generalization, preserving LVLM's other foundational capabilities.

## 2 ANALYSIS OF CAPTION QUERIES' EFFECT ON VISUAL ATTENTION

We performed a quantitative analysis to validate the primary motivation for CAI: caption queries uniquely refine visual attention patterns in LVLMs in a way that other queries do not. Using a sample of 1,000 images from the MS-COCO dataset (Lin et al., 2014), we designed three distinct queries for each image to analyze the effect of query type: one caption query and two vision-oriented non-caption queries with distinct meanings (non-caption-1 & non-caption-2). To quantify the effect on visual attention for caption and non-caption queries, we compute the **Change Rate** of attention weights across all layers and attention heads. Further details on this computation and the experimental setup are available in Appendix A.

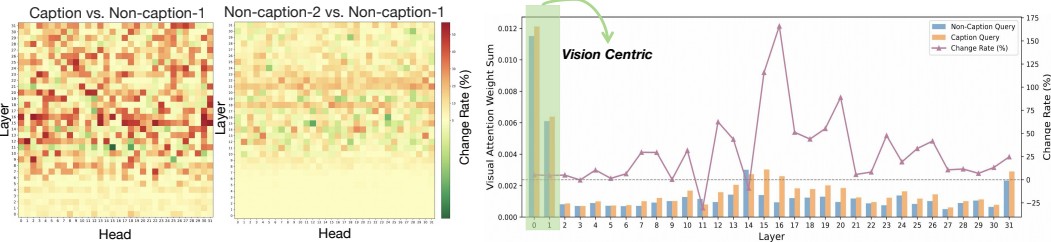

(a) Head-wise Visual Attention Weight Change Rate    (b) Layer-wise Visual Attention Weight Sum&Change Rate

Figure 2: A quantitative analysis from head-wise (a) and layer-wise (b) perspective on visual attention weights, which demonstrates that caption queries significantly enhance visual attention of LLaVA-1.5-7b.

Experimental results in Figure 2 indicate that caption queries demonstrate significant enhancements on LVLM's visual attention weights compared with non-caption queries, especially in the mid layers. As shown in Figure 2 (a), 65.92% of attention heads, which are concentrated primarily in middle layers, exhibit increased visual attention weights when fed caption queries. As shown in Figure 2 (b), 30 out of 32 layers exhibit a consistent enhancement in visual attention. Notably, the mid-layer attention heads demonstrate the most substantial improvements, which indicates their critical role in enabling LVLMs' fine-grained perception capability. Our analysis provides clear feasibility and insights for locating and refining attention heads by leveraging the visual attention enhancement induced by caption queries to mitigate object hallucinations.

## 3 METHODS

### 3.1 PRELIMINARIES: THE TRANSFORMER RESIDUAL STREAM

We consider a LVLM parametrized by $\theta$. The model receives as input a visual input $\boldsymbol{V} = \{v_1, v_2, \ldots, v_m\}$ and a textual query $\boldsymbol{T} = \{t_1, t_2, \ldots, t_n\}$, where $m$ and $n$ denote the sequence lengths of the visual input and textual inputs. The textual and visual inputs are concatenated together to form the first layer input $\boldsymbol{H}^1 = \text{concat}(\boldsymbol{V}, \boldsymbol{T}) \in \mathbb{R}^{(m+n) \times d}$ for the $L$ layers $\times H$ heads language decoder.

During the forward pass, the input $\boldsymbol{H}^l$ received by the $h$-th attention head at $l$-th layer is linearly transformed using independent weight matrices to generate the Query, Key and Value matrices, denoted as $\boldsymbol{Q}_{(l,h)} \in \mathbb{R}^{(m+n) \times d}$, $\boldsymbol{K}_{(l,h)} \in \mathbb{R}^{(m+n) \times d}$ and $\boldsymbol{V}_{(l,h)} \in \mathbb{R}^{(m+n) \times d}$, where $d$ denotes the head-specific hidden dimension. The generated Query, Key, and Value matrices are then used to compute the attention score, attention weight matrix, and attention output as follows:

$$\dot{\boldsymbol{A}}_{(l,h)} = \frac{\boldsymbol{Q}_{(l,h)} \boldsymbol{K}_{(l,h)}^T}{\sqrt{d}}, \boldsymbol{A}_{(l,h)} = \text{softmax}(\dot{\boldsymbol{A}}_{(l,h)} + \boldsymbol{M}), \boldsymbol{M}[i,j] = \begin{cases} 0 & \text{if } j \leq i \\ -\infty & \text{if } j > i \end{cases} \quad (1)$$

$$\boldsymbol{O}_{(l,h)} = \boldsymbol{A}_{(l,h)} \boldsymbol{V}_{(l,h)}, \quad (2)$$

where $\boldsymbol{M}$ is the causal mask matrix. At each layer, the hidden states pass through multi-head attention (MHA), which comprise $H$ independent attention heads, each performing separate linear transformations. Specifically, the MHA mechanism can be formulated as:

$$\boldsymbol{H}^{l+1} = \boldsymbol{H}^l + \sum_{h=1}^{H} \boldsymbol{O}_{(l,h)} \cdot \boldsymbol{W}_o^l, \quad (3)$$

where $\boldsymbol{W}_o^l \in \mathbb{R}^{Hd \times d}$ is the learnable weight matrix and maps d-dimensional attention outputs of heads into hidden state representations, which are then fed into a standard multilayer perception (MLP) for further processing. Finally, the model predicts the next token in auto-regressive manner.

### 3.2 CAPTION-SENSITIVE ATTENTION HEADS PROBE

This module aims to identify caption-sensitive attention heads, which are also visually sensitive and exhibit significant differences in attention outputs when responding to caption and non-caption

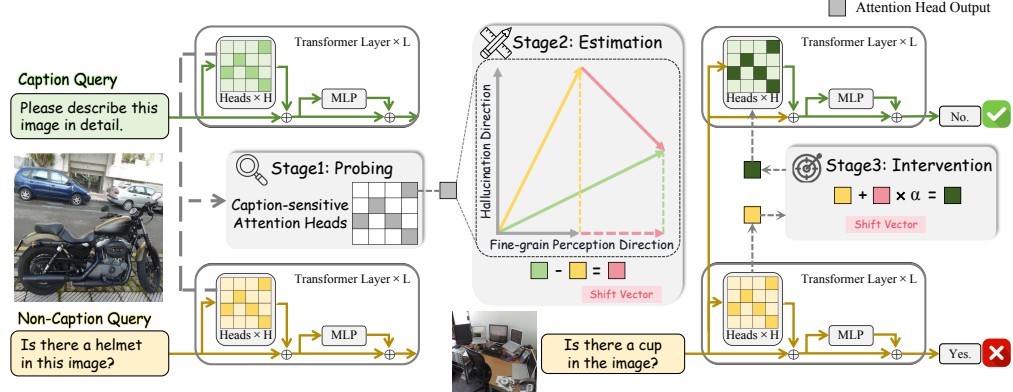

Figure 3: An overview of the CAI method. Each square in the matrix represents the attention head output. Squares with dark green color indicate refined caption-sensitive attention head outputs. CAI consists of three stages: **(1) Caption-Sensitive Attention Heads Probe §3.2:** We use probing techniques to identify caption-sensitive attention heads, which exhibit enhanced visual attention when fed caption queries versus non-caption queries. **(2) Estimation of Perception Refined Vectors §3.3:** We estimate the perception refined vectors by computing the attention output shift vectors from feeding non-caption queries to caption queries. **(3) Intervention at Inference Time §3.4:** We apply the precomputed attention refined vectors to the Top-$K$ caption-sensitive attention heads during inference, thereby enhancing visual attention and activating the model's inherent fine-grained visual perception capability and effectively mitigate object hallucination.

queries. Since LVLMs generate tokens in an auto-regressive manner, CAI focuses on the attention matrices of the last input token, $\dot{\boldsymbol{A}}_{(l,h)}[m+n]$, which aggregates the most comprehensive visual and textual information. Furthermore, we aim to capture the differences in attention activation patterns when fed caption queries versus non-caption queries, as well as minimize the influence of textual semantic information during probing. To achieve this, we mask $\dot{\boldsymbol{A}}_{(l,h)}[m+n]$ to exclude attention towards all textual tokens during the forward pass, and compute the modified attention output:

$$\hat{\boldsymbol{M}}[i,j] = \begin{cases} 0 & \text{if } j \leq i \\ -\infty & \text{if } j > i \text{ or } (i = m+n \text{ and } j > n) \end{cases} \tag{4}$$

$$\hat{\boldsymbol{O}}_{(l,h)} = \text{softmax}(\dot{\boldsymbol{A}}_{(l,h)} + \hat{\boldsymbol{M}})\boldsymbol{V}_{(l,h)}, \widetilde{\boldsymbol{O}}_{(l,h)} = \hat{\boldsymbol{O}}_{(l,h)}[m+n]. \tag{5}$$

For a dataset with a batchsize of $B$, the last token's modified attention output of $b$-th VQA problem when answering caption query and non-caption query are denoted as $\widetilde{\boldsymbol{O}}^b_{(l,h)}$ and $\widetilde{\boldsymbol{O}'}^b_{(l,h)}$. For each attention head $Head_{(l,h)}$, we use $B$ pairs of modified attention output to train a binary classifier $f_{l,h}(\cdot)$ that predicts whether the input sentence is a caption query. Finally, we select the attention heads with the Top-$K$ highest classification accuracy as the caption-sensitive attention heads. The formulas are summarized as:

$$f^*_{l,h} = \underset{f_{l,h}(\cdot)}{\arg\min} \sum_{b=1}^{B} \mathcal{L}\left(f_{l,h}\left(x_b\right), y_b\right), \tag{6}$$

$$Heads = \{Head_{(l,h)} \mid Head_{(l,h)} \in \text{TopK}(\text{Acc}(f^*_{l,h}))\} \tag{7}$$

where $f^*_{l,h}$ denotes the final probe, $\mathcal{L}$ denotes the loss function of the probes, $x_b \in \{\widetilde{\boldsymbol{O}}^b_{(l,h)}, \widetilde{\boldsymbol{O}'}^b_{(l,h)}\}$ denotes the input of the classifier, $y_b \in \{0, 1\}$ denotes the category of query (0 for caption query, 1 for non-caption query, respectively), and $K$ denotes the number of selected heads.

### 3.3 ESTIMATION OF PERCEPTION REFINED VECTORS

This module aims to use caption-sensitive attention heads to accurately estimate the perception refined vectors. For a dataset with a batchsize of $B$, the last token's origin attention output of $b-$th

VQA problem when answering caption query and non-caption query are denoted as $\boldsymbol{O}^b_{(l,h)}$ and $\boldsymbol{O'}^b_{(l,h)}$. To estimate the fine-grained perception direction for each attention head, attention output shift vector is computed as follows:

$$\boldsymbol{S}_{(l,h)} = \frac{1}{B} \sum_{b=1}^{B} \left( \boldsymbol{O}^b_{(l,h)} - \boldsymbol{O'}^b_{(l,h)} \right).$$ (8)

These shift vectors estimate the visual attention difference between caption queries and non-caption queries, which serve as the fine-grained perception directions. In particular, the modified attention outputs $\widetilde{\boldsymbol{O}}^b_{(l,h)}, \widetilde{\boldsymbol{O'}}^b_{(l,h)}$ are not used to estimate the refined vectors, as these values are not directly derived from the original inference process. In contrast, using the original attention outputs leads to more robust refined vectors.

### 3.4 Intervention at Inference Time

This module aims to refine caption-sensitive attention heads at inference time. We leverage the precomputed refined vectors to steer these heads from insufficient visual attention states to sufficient states, thereby enhancing the model's fine-grained visual perception capability and mitigate hallucination. At each layer, the updated hidden state after intervention is computed as:

$$\boldsymbol{H}^{l+1} = \boldsymbol{H}^l + \sum_{h=1}^{H} \left( \boldsymbol{O}_{(l,h)} + \mathbb{I}_{(l,h)} \alpha \boldsymbol{S}_{(l,h)} \right) \cdot \boldsymbol{W}^l_o,$$ (9)

where $\mathbb{I}_{(l,h)}$ is a gating function, assigning a value of 1 to caption-sensitive attention heads, and 0 to the others. $\alpha$ represents the intensity of the intervention.

In conclusion, CAI significantly enhances LVLM's fine-grained perception capability, which is attributed to the unique role of caption queries during the pre-training stage for text-image alignment, and their sufficient visual attention patterns. Furthermore, CAI benefits from the inference-time intervention paradigm, which provides an inherent advantage in inference latency.

## 4 Experiments

### 4.1 Experimental Setup

**Benchmarks.** We evaluate our proposed CAI method across five benchmarks, including discriminative and generative tasks to measure its effectiveness and robustness. See Appendix B.1 for details of benchmarks.

**Baselines.** We adopt LLaVA-1.5-7b, Qwen-VL-Chat, LLaVA-NeXT (Liu et al., 2024a) as baseline LVLMs, compared with several SOTA training-free methods. See Appendix C for results on more advanced LVLMs, and Appendix D for results compared with other SOTA training-free methods.

**(1) Baselines tailored for decoding:** VCD (Leng et al., 2024) contrasts model logits derived from original and distorted visual input to reduce the over-reliance on statistical bias. OPERA (Huang et al., 2024) introduces a penalty term on the logits during the beam-search decoding to mitigate the over-trust issue.

**(2) Baselines utilizing inference-time intervention (ITI):** PAI (Liu et al., 2024c) intervenes on attention heads by leveraging their original direction and optimizes the output distribution during decoding to mitigate language bias. VTI (Liu et al., 2024b) mitigates hallucination by steering hidden states at inference time to enhance the stability of visual features.

Despite prior findings (Bi et al., 2024b) indicating the significant role of attention heads in visual perception, there is a lack of approaches that analyze at head level and do not rely on specific decoding strategies (which increase inference time). The idea of using the attention differential between caption and non-caption inputs to guide inference interventions distinguishes CAI from earlier ITI works.

**Implementation Details.** In our experiments, we utilize 1000 task-diverse VQAs from LLaVA-1.5-7b pretraining dataset, each paired with a specific caption query, to identify caption-sensitive

attention heads and compute the attention shift vectors. For each attention head, SVM (Cortes, 1995) is used as the classifier and two-fold cross-validation is performed to evaluate its accuracy. More details are provided in Appendix B.

## 4.2 MAIN RESULTS

As shown in Figure 4 and Table 1-3, we summarize our main findings as follows:

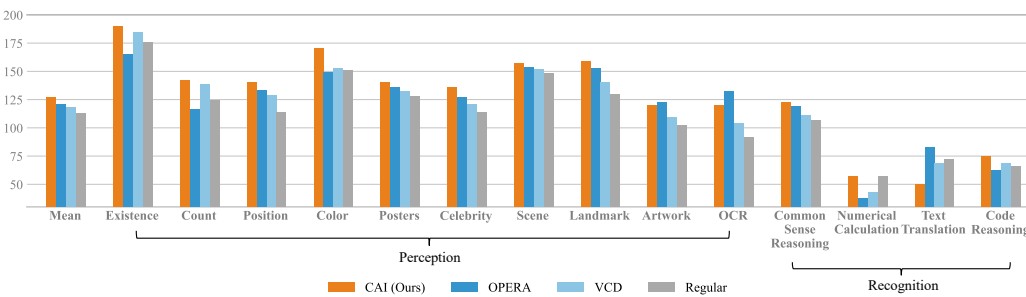

Figure 4: Main results of LLaVA-1.5-7b on the MME.

| Setting | Method | LLaVA-1.5-7b | | Qwen-VL-Chat | | LLaVA-NeXT | |
|---|---|---|---|---|---|---|---|
| | | Accuracy↑ | F1-Score↑ | Accuracy↑ | F1-Score↑ | Accuracy↑ | F1-Score↑ |
| Random | Regular | 83.29 | 81.33 | 84.63 | 82.61 | 84.78 | 86.43 |
| | VCD | 87.73 | 87.16 | 86.93 | 85.46 | 88.76 | 89.57 |
| | OPERA | 89.20 | 88.81 | 85.71 | 84.64 | 90.27 | 89.71 |
| | PAI | 86.33 | 84.56 | 85.38 | 85.54 | 88.40 | 87.16 |
| | VTI | 89.50 | 88.89 | 86.73 | 85.59 | 89.23 | 88.68 |
| | CAI(ours) | 89.87 (+6.58) | 89.43 (+8.10) | 88.17 (+3.54) | 87.31 (+4.70) | 90.68 (+5.90) | 90.42 (+3.99) |
| Popular | Regular | 81.88 | 80.06 | 83.63 | 81.53 | 83.23 | 84.77 |
| | VCD | 85.38 | 85.06 | 85.17 | 83.68 | 87.01 | 87.70 |
| | OPERA | 86.64 | 86.62 | 84.82 | 83.99 | 87.16 | 87.68 |
| | PAI | 85.33 | 83.62 | 84.20 | 83.10 | 86.65 | 86.99 |
| | VTI | 87.36 | 86.69 | 85.67 | 84.48 | 87.33 | 87.16 |
| | CAI(ours) | 88.32 (+6.44) | 87.95 (+7.89) | 87.73 (+4.10) | 86.84 (+5.31) | 89.53 (+6.30) | 89.24 (+4.47) |
| Adversarial | Regular | 78.96 | 77.57 | 81.03 | 79.30 | 81.19 | 82.50 |
| | VCD | 80.88 | 81.33 | 83.10 | 82.04 | 84.80 | 85.23 |
| | OPERA | 81.24 | 81.38 | 82.67 | 79.89 | 85.20 | 85.54 |
| | PAI | 83.17 | 81.67 | 82.19 | 82.06 | 84.32 | 83.68 |
| | VTI | 82.57 | 82.11 | 83.13 | 82.16 | 85.35 | 84.52 |
| | CAI(ours) | 84.27 (+5.31) | 84.41 (+6.84) | 84.33 (+3.30) | 83.92 (+4.62) | 85.97 (+4.78) | 86.07 (+3.57) |

Table 1: Main results on POPE tasks. The best performances are bolded.

| Method | LLaVA-1.5-7b | | | | Qwen-VL-Chat | | | |
|---|---|---|---|---|---|---|---|---|
| | $C_S$↓ | $C_I$↓ | Recall↑ | Len | $C_S$↓ | $C_I$↓ | Recall↑ | Len |
| Regular | 52.8 | 15.9 | 77.3 | 93.4 | 2.8 | 3.0 | 31.0 | 5.3 |
| VCD | 51.0 | 14.9 | 77.2 | 101.9 | 1.4 | 1.2 | 30.8 | 4.0 |
| OPERA | 45.6 | 13.1 | **78.5** | 95.3 | 1.7 | 1.3 | 31.9 | 4.4 |
| PAI | 38.3 | 12.4 | 76.9 | 94.4 | 1.3 | 1.2 | 32.2 | 4.2 |
| VTI | 36.9 | 12.1 | 76.8 | 93.8 | 1.1 | 1.1 | 31.4 | 4.2 |
| CAI | **34.6** | **11.5** | 78.2 | 95.8 | **1.0** | **0.9** | **32.6** | 4.4 |

Table 2: Results on CHAIR benchmark. Max new tokens are set to be 512.

| Method | LLaVA-1.5-7b | | | Qwen-VL-Chat | | |
|---|---|---|---|---|---|---|
| | Score↑ | VH.%↓ | Hu.%↓ | Score↑ | VH.%↓ | Hu.%↓ |
| Regular | 1.86 | 63.5 | 67.1 | 2.93 | 41.1 | 61.0 |
| VCD | 2.12 | 54.2 | 66.7 | 2.77 | 39.2 | 61.5 |
| OPERA | 2.15 | 54.2 | 63.0 | 2.94 | 38.4 | 58.2 |
| PAI | 2.27 | 53.2 | 62.5 | 2.87 | 39.5 | 56.7 |
| VTI | 2.33 | 52.2 | 63.4 | 2.99 | 38.4 | 57.4 |
| CAI | **2.43** | **51.0** | **61.5** | **3.04** | **38.0** | **56.0** |

Table 3: Results on MMHal-Bench and MHumanEval (evaluated by GPT-4 & Human).

**(1) SOTA hallucination mitigation performance** Our proposed CAI method achieves SOTA hallucination mitigation performance across both discriminative and generative tasks. On the POPE benchmark, CAI improves accuracy by an average of +5.64% and F1 Score by +5.50%. On the CHAIR benchmark, CAI reduces the average hallucination metrics ($C_S$ and $C_I$) by 6.43 points. On MMHal-Bench, CAI improves the average Score by +0.16, while reduces the average VH Rate by 2.95% and the Hu. Rate by 2.25%. As shown in Appendix L , CAI substantially mitigates the "yes-bias", providing deeper evidence of CAI's effectiveness in discriminative settings.

**(2) Generalizability across architectures and datasets** CAI exhibits strong generalization capability across both model architectures and data sources. From the architectural perspective, CAI remains effective across models with different attention mechanisms, including those with optimized implementations such as Qwen-VL-Chat. This is because CAI stems from the difference in attention patterns between caption and non-caption queries, rather than the specific implementation details of the multi-head attention mechanism. From the data perspective, although the probing and refined vectors are computed using 1,000 samples from the LLaVA-1.5-7b pre-training dataset, they generalize well to other out-of-domain benchmarks and advanced LVLMs. These results highlight the generalizability across model architectures and datasets.

**(3) Preservation of foundational capabilities** CAI not only mitigates hallucination but also preserves the LVLM's other foundational capabilities. On the MME benchmark, CAI improves performance on 13 out of 14 tasks, preserving most of LVLM's foundational capabilities. Furthermore, CAI improves the informativeness score by 0.16 on MMHal-Bench, demonstrating that CAI effectively mitigates object hallucination without compromising informativeness.

## 5 ANALYSIS AND DISCUSSIONS

### 5.1 OPTIMIZATION VIA CAPTION QUERIES' DIVERSITY

| Setting | VA (%) | Parameters | | Random | | Popular | | Adversarial | | Average | |
|---|---|---|---|---|---|---|---|---|---|---|---|
| | | $\alpha$ | $K$ | ACC↑ | F1↑ | ACC↑ | F1↑ | ACC↑ | F1↑ | ACC↑ | F1↑ |
| Regular | 31.4 | - | - | 83.29 | 81.33 | 81.88 | 80.06 | 78.96 | 77.57 | 81.38 | 79.65 |
| Random1 | 46.8 (+15.4) | 1.25 | 100 | 88.59 | 88.15 | 86.95 | 86.55 | 83.08 | 83.25 | 86.21 | 85.98 |
| Random2 | 45.6 (+14.2) | 1.50 | 100 | 88.65 | 88.21 | 87.01 | 86.68 | 83.15 | 83.33 | 86.27 | 86.07 |
| Random3 | 44.7 (+13.3) | 1.50 | 125 | 89.02 | 88.65 | 87.41 | 87.05 | 83.58 | 83.72 | 86.67 | 86.47 |
| Random4 | 44.2 (+12.8) | 1.50 | 100 | 89.15 | 88.82 | 87.53 | 87.21 | 83.66 | 83.80 | 86.78 | 86.61 |
| Optimized of $N$ | 43.4 (+12.0) | 1.50 | 100 | **89.87** | **89.43** | **88.32** | **87.92** | **84.27** | **84.41** | **87.49** | **87.26** |
| Ensemble of $N$ | - | 1.50 | 100 | 88.93 | 88.68 | 87.46 | 86.91 | 83.78 | 83.56 | 86.72 | 86.38 |

Table 4: We construct a caption query candidate pool (N=16), where we derive our test cases as follows: (1) four queries are randomly selected; (2) one optimal query is chosen using caption query optimization algorithm; and (3) an ensemble intervention strategy is applied. **VA** (%) indicates the average percentage of attention weights over visual tokens when fed corresponding query. $\alpha$ and $K$ denote the intensity and number of the intervention. We select the optimal parameters separately for each setting.

To further enhance the robustness of CAI, we aim to leverage the diversity of caption queries and introduce two optimization strategies to improve real-world application.

**Candidate Caption Query Pool Expansion:** Caption queries refer to prompts with explicit semantics (e.g., "Please describe this image in detail") and strong cross-model transferability, which can be easily sourced from open pre-training datasets or generated using large language models. By expanding the candidate pool, we increase the diversity and generalizability of caption-sensitive attention heads probing.

**Caption Query Optimization Algorithm:** Our experiments reveal that the shift cost—the attention weights change from a non-caption query to a caption query on a dataset—varies when fed different caption queries. Caption queries with minimal necessary shift cost yield better hallucination mitigation performance and we term these queries as optimized queries. This is possibly because optimized queries require less attention diversion from textual to visual information while still en-

abling fine-grained perception capability. As a result, optimized queries preserve the model's native attention distribution better and strike a balance between visual and textual attention. As shown in Table 4, by expanding the pool of candidate caption queries and applying the proposed caption query optimization algorithm, we can further enhance CAI's performance.

**Multi-query Feature Ensemble Algorithm:** Although CAI achieves stable performance across different caption queries, we propose a multi-query ensemble strategy to reduce the influence of sub-optimal or outlier queries. Specifically, we integrate attention features from multiple caption queries to identify consistent caption-sensitive heads and estimate perception refined vectors. Strengthening these heads improves object hallucination mitigating performance and provides robust intervention against individual prompt variability. As shown in Table 4, while this ensemble may be marginally less optimal than using the optimized caption query, it substantially improves the reliability of CAI under various conditions.

## 5.2 DISTRIBUTION OF CAPTION-SENSITIVE ATTENTION HEADS

As illustrated in Figure 5, we visualize the classification accuracies across $32 \times 32$ attention heads during the probing stage of LLaVA-1.5-7B (left) and Qwen-VL-Chat (right). We observe that caption-sensitive attention heads are concentrated primarily between the 7th and 20th layers, which is well aligned with the layers with higher **Change Rat**es presented in Figure 2. These attention heads play a critical role to fine-grained visual perception. By refining the output of these heads, CAI effectively enhances LVLM's visual perception capability and mitigates object hallucination.

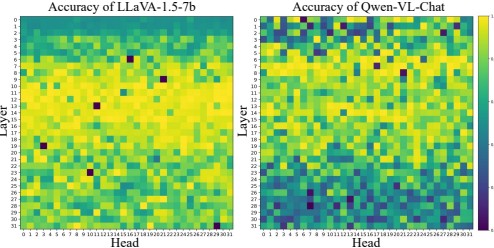

Figure 5: The accuracies of probes.

Figure 6: Ablation study of $\alpha$ and $K$ on POPE.

| Method | $C_S \downarrow$ $C_I \downarrow$ | PPL | Coher. ↑ | Fluency ↑ |
|---|---|---|---|---|
| LLaVA-1.5-7b | 20.80 6.77 | 3.97 | 0.998 | 0.805 |
| + CAI | 17.20 5.50 | 4.11 | 0.998 | 0.791 |
| + CAI (over) | 18.60 6.00 | 4.23 | 0.997 | 0.809 |

Table 5: Impact of over-intervention on CHAIR benchmark. $Max\ new\ tokens$ is set to 64.

| Method | TTFT(ms) | TPOT(ms) | Acc(%) |
|---|---|---|---|
| LLaVA-1.5-7b | 99.8 1.0× | 36.0 1.0× | 78.96 |
| + VCD | 160.1 1.6× | 96.8 2.7× | 80.88 |
| + PAI | 156.3 1.6× | 93.6 2.6× | 83.17 |
| + CAI(ours) | 102.2 1.0× | 36.5 1.0× | 84.50 |

Table 6: Inference latency (TTFT, TPOT) and accuracy on POPE adversarial.

## 5.3 IMPACT OF HYPERPARAMETERS AND INFERENCE LATENCY

CAI method primarily relies on two key hyperparameters: the intensity of intervention $\alpha$ and the number of refined attention heads $K$. We use grid search to find the optimal value for both hyperparameters across benchmarks. See Appendix I for detailed results. As shown in Figure 6, we provide the ablation study results for each hyperparameter when the other is fixed to its optimal value ($K = 100$ on the left and $\alpha = 1.5$). The key implications can be summarized as follows:

(1) Impact of $\alpha$: When $\alpha$ is small, the attention intervention is insufficient, leading to marginal improvements. While a large $\alpha$ leads to insufficient attention to textual information, leading to a performance drop.

(2) Impact of $K$: Applying intervention to few attention heads fails to influence the full activation pathways of visual information. While intervening in excess heads disrupts attention activation paths that are irrelevant to visual perception and play essential roles in other foundational capabilities, leading to performance drop.

Moreover, as shown in Table 5, we employ UniEval (Zhong et al., 2022) and perplexity (PPL) computation to evaluate the coherence and fluency of generated responses. We find even when doubling the optimal intervention parameter, CAI does not compromise the coherence and fluency of outputs. Furthermore, as shown in Table 6, CAI achieves better hallucination mitigating performance with less additional inference latency, which benefits from the inference-time intervention paradigm.

## 5.4 CASE STUDY

CAI remains effective in caption task, which is attributed to the enhancement in visual attention. As shown in Figure 7, CAI effectively mitigates object hallucination not only during the regeneration of new responses, but also when extending hallucinated contexts, highlighting its fine-grained, token-level object hallucination mitigation capability.

## 6 RELATED WORKS

### 6.1 LARGE VISION-LANGUAGE MODELS

Several powerful LVLMs based on open-source LLM backbones combined with visual encoders have achieved impressive capabilities through vision-language pretraining. Furthermore, recent searches have further improved model performance by employing high-resolution visual encoders (Hong et al., 2024) and exploring reinforcement learning methods, such as RLHF (Yu et al., 2024a). Closed-source models, such as GPT-4o (Hurst et al., 2024) and Gemini 1.5 (Reid et al., 2024) have demonstrated even more powerful performance. In addition, a growing body of work emphasizes scaling strategies, cross-modal alignment, and integration of external knowledge sources, which further enrich the reasoning and generation abilities of LVLMs. However, despite these advances, recent LVLMs still suffer from hallucination problems, and addressing how to cost-effectively mitigate hallucination remains an important open question that demands deeper exploration.

### 6.2 MITIGATING HALLUCINATION IN LVLMS

Current methods for mitigating hallucination in LVLMs can be broadly categorized into two types: data-driven training methods and training-free methods. Training-based methods typically involve introducing novel training objectives (Chen et al., 2024a) and utilizing carefully curated datasets (Gunjal et al., 2024; Liu et al., 2023b; Yu et al., 2024b; You et al., 2023). For training-free methods, the main strategies include designing decoding techniques (Leng et al., 2024; Chen et al., 2024b; Chuang et al., 2023; Huang et al., 2024; Zhong et al., 2024) during the inference phase and leveraging language or visual prompts (Lee et al., 2023; An et al., 2024). PAI (Liu et al., 2024c) intervenes in attention heads by leveraging the direction and magnitude of their original outputs, and optimizes the output distribution during decoding to mitigate hallucinations. VTI (Liu et al., 2024b) reduces hallucinations by steering hidden states during inference to enhance the stability of vision features. Beyond these approaches, a number of studies highlight the importance of understanding the underlying mechanisms that trigger hallucinations, suggesting that architectural and interpretability-driven interventions may offer complementary solutions. However, our work is the first to explicitly reveal the impact of caption queries on the attention activation patterns of LVLMs and mitigate hallucination by applying caption-sensitive attention head intervention during inference.

## 7 CONCLUSION

In this paper, we are the first to explicitly reveal the impact of caption queries versus non-caption queries on the attention activation patterns of LVLMs, providing novel insights for the optimization of visual attention. Furthermore, we propose CAI, a training-free method that probes and refines caption-sensitive attention heads during inference, thereby enhancing LVLM's fine-grained perception capabilities and mitigating object hallucination. Comprehensive experimental results across five widely used benchmarks demonstrate that CAI not only effectively mitigates hallucination with little inference latency, but also shows strong generalization, preserving foundational capabilities.

## REPRODUCIBILITY STATEMENT

We are committed to ensuring the reproducibility of our work. All datasets and models used in our work are publicly available, as noted in Appendix B.1. The detailed experimental settings, parameters and more results are provided in Appendix B, C and D.

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

## A    EXPERIMENTAL SETUP OF QUANTITATIVE ANALYSIS

We sample 1,000 images from the MS-COCO dataset (Lin et al., 2014). For each image, we propose one caption query and two different non-caption queries (non-caption-1 & non-caption-2) to analyze differences attributable to query types.

We consider a LVLM parametrized by $\theta$. The model receives as input a textual query $\boldsymbol{T} = \{t_1, t_2, \ldots, t_n\}$ and a visual input $\boldsymbol{V} = \{v_1, v_2, \ldots, v_m\}$, where $n$ and $m$ denote the sequence lengths of the text and visual inputs. The text and vision inputs are concatenated together to form the first layer input $\boldsymbol{H}^1 = \mathrm{concat}(\boldsymbol{V}, \boldsymbol{T}) \in \mathbb{R}^{(m+n) \times d}$ for the $L$ layers $\times$ $H$ heads decoder. For an image, the last input token's visual attention weight of $H$-th head in $L$-th layer $\boldsymbol{Sum}_{(l,h)}$ can be computed as:

$$\boldsymbol{A}_{(l,h)} = \mathrm{softmax}(\frac{\boldsymbol{Q}_{(l,h)}\boldsymbol{K}_{(l,h)}^T}{\sqrt{d}}), \tag{10}$$

$$\boldsymbol{Sum}_{(l,h)} = \sum_{i=1}^{m} \boldsymbol{A}_{(l,h)}^{-1}[i], \tag{11}$$

where the $\boldsymbol{Q}_{(l,h)}$ and $\boldsymbol{K}_{(l,h)}$ are the Query and Key matrixs of the $k$-th head in $l$-th layer, $\boldsymbol{A}_{(l,h)}^{-1}[i]$ is the last input token's attention weight of the $i$-th input token. For a dataset of $B$ samples, the sum of visual attention weight can be computed as:

$$S_{(l,h)} = \sum_{b=1}^{B} \boldsymbol{Sum}_{(l,h)}. \tag{12}$$

Then we record the sum of visual attention weights from the last input token for three types of queries: $S_{(l,h)}^{cap}$ for caption query, $S_{(l,h)}^{non-1}$ for non-caption query 1 and $S_{(l,h)}^{non-2}$ for non-caption query 2. The head-wise Change Rate $Rate_{(l,h)}$ and layer-wise Change Rate $Rate_{(l)}$ can be computed as:

$$Rate_{(l,h)}^{cap} = \frac{S_{(l,h)}^{cap} - S_{(l,h)}^{non-1}}{S_{(l,h)}^{non-1}}, Rate_{(l,h)}^{non-cap} = \frac{S_{(l,h)}^{non-2} - S_{(l,h)}^{non-1}}{S_{(l,h)}^{non-1}}, \tag{13}$$

$$Rate_{(l)}^{cap} = \frac{\sum_{h=1}^{H}(S_{(l,h)}^{cap} - S_{(l,h)}^{non-1})}{\sum_{h=1}^{H} S_{(l,h)}^{non-1}}, Rate_{(l)}^{non-cap} = \frac{\sum_{h=1}^{H}(S_{(l,h)}^{non-2} - S_{(l,h)}^{non-1})}{\sum_{h=1}^{H} S_{(l,h)}^{non-1}}. \tag{14}$$

By comparison, we find that visual attention across particular attention heads was significantly enhanced when fed caption compared to non-caption queries. These results provide strong support for our proposed motivation.

## B    ADDITIONAL EXPERIMENTAL DETAILS

### B.1    BENCHMARKS

We evaluate our proposed CAI method across five benchmarks, including both discriminative and generative tasks to measure its effectiveness and robustness:

**(1) POPE** (Li et al., 2023b) employs a binary question-answering format, inquiring LVLMs to answer if a special object exists in the given image. We adopt Accuracy and F1 score as the evaluation metrics.

**(2) MME** (Fu et al., 2023) serves as a comprehensive tool for assessing the capabilities of LVLMs across 10 perception tasks and 4 cognition tasks. Consequently, task scores are reported as the evaluation metrics.

**(3) CHAIR** (Rohrbach et al., 2018) is a widely used metric to assess object hallucination of LVLMs. The CHAIR metric comprises two indicators, denoted as $C_S$ and $C_I$, with the following calculation formulas:

$$C_S = \frac{|\{\text{Hallucinated objects}\}|}{|\{\text{All mentioned objects}\}|}$$

$$C_I = \frac{|\{\text{Sentences w/ hallucinated objects}\}|}{|\{\text{All sentences}\}|}$$

**(4) MMHal-Bench** (Sun et al., 2023) comprises 96 meticulously designed questions, which evaluates response-level hallucination rate (VH.%) and informativeness (Score). It asks **GPT-4** (Achiam et al., 2023) to compare model outputs with human responses and object labels for evaluation.

**(5) MHumanEval** (Yu et al., 2024b) is designed to evaluate hallucination performance by **human annotators**. The benchmark contains 146 samples collected from Object HalBench and MMHal-Bench. Given model responses, we ask three human annotators to label the hallucinated segments and compute the mean response-level hallucination rate (Hu.%) as the evaluation metric.

## B.2  DATA SOURCE

Although our method does not rely on specific data, we separately specify the sources of the data used in the experiments for the sake of reproducibility.

### B.2.1  DATA OF BEST QUERY SEARCH

In the best caption search algorithm, we use the top 100 VQA samples from the complex reasoning data in the LLaVA-1.5-7b pre-training dataset. From this, we obtain non-caption queries and their corresponding images. Additionally, we maintain a list of 16 candidate caption queries, some of which are manually generated and others are derived from the pre-trained instructions of LLaVA-1.5-7b. The caption query candidates are listed as follows:

*"What do you see happening in this image?"*, *"What do you think is going on in this snapshot?"*, *"Can you elaborate on the elements of the picture provided?"*, *"Describe the following image."*, *"What's happening in the scene?"*, *"Analyze the image in a comprehensive and detailed manner."*, *"Write a detailed description of the given image."*, *"What is this photo about?"*, *"Explain the visual content of the image in great detail."*, *"What are the key elements in this picture?"*, *"Can you describe the main features of this image for me?"*, *"Please describe this image in detail."*, *"Generate the caption in English:"* *"Provide a thorough narrative of what the image depicts."* *"Offer a detailed explanation of the scene captured in the picture."* *"Summarize the visual information conveyed by this image."*

In the experiments, the best caption query for LLaVA-1.5-7b and LLaVA-NeXT is *"Analyze the image in a comprehensive and detailed manner."* and the best caption query for Qwen-VL-Chat, InternVL2-8B, Qwen2-VL-7B and Qwen2.5-VL-7B is *"Please describe this image in detail."*.

### B.2.2  DATA OF PROBE AND SHIFT COMPUTATION

We extracted the first 1,000 samples from the complex reasoning data in the LLaVA-1.5-7b pre-training dataset. The questions from these samples were treated as non-caption queries.

## B.3  DETAILED EXPERIMENTAL SETUP

In the experiment of POPE, 'regular' refers to the direct sampling setting. We used direct sampling decoding and set $\alpha = 1.5$ and $K = 100$ in the main experiments.

## C  RESULTS ON MORE ADVANCED MODELS

As shown in Table 7, CAI further exhibits effective hallucination mitigation when applied to more advanced models, providing additional evidence for the generalizability of CAI.

| Model | POPE | | MME | | CHAIR | | MMHal-Bench | | |
|---|---|---|---|---|---|---|---|---|---|
| | Acc(%) ↑ | F1-Score(%) ↑ | Cog.↑ | Hall.%↑ | $C_S$ ↓ | $C_I$ ↓ | Score↑ | VH.%↓ | Hu.%↓ |
| Qwen2-VL-7B | 88.49 | 87.85 | 556.4 | 630.0 | 24.8 | 7.2 | 2.87 | 49.8 | 55.4 |
| + CAI | **89.85** | **89.87** | **570.4** | **668.3** | **15.6** | **6.5** | **3.09** | **40.8** | **48.4** |
| InternVL2-8B | 86.67 | 85.72 | 566.4 | 663.0 | 37.2 | 9.4 | 2.71 | 52.3 | 56.7 |
| + CAI | **87.98** | **87.42** | **573.3** | **693.7** | **31.3** | **8.4** | **2.91** | **44.4** | **49.7** |
| LLaVA-NeXT | 83.06 | 84.57 | 533.7 | 586.7 | 40.0 | 10.5 | 2.57 | 55.8 | 65.4 |
| + CAI | **88.73** | **88.58** | **566.7** | **657.5** | **33.3** | **8.9** | **3.12** | **48.9** | **61.0** |
| Qwen2.5-VL-7B | 87.35 | 87.09 | 630.0 | 683.3 | 37.2 | 8.7 | 3.05 | 34.7 | 43.6 |
| + CAI | **88.96** | **88.70** | **655.7** | **695.0** | **32.6** | **8.0** | **3.24** | **29.9** | **40.2** |

Table 7: Results on more advanced LVLMs, including Qwen2-VL-7B (Wang et al., 2024a), InternVL2-8B (Chen et al., 2024c), LLaVA-NeXT and Qwen2.5-VL-7B (Bai et al., 2025). Cog. and Hall. denote the cognitive and hallucination subset of MME benchmark.

## D  COMPARISON WITH MORE ADVANCED METHODS

We selected LLaVA-1.5-7b as the baseline model and compared CAI with more advanced models including VCD (Leng et al., 2024), ICD (Wang et al., 2024b), OPERA (Huang et al., 2024), Woodpecker (Yin et al., 2024), M3ID (Favero et al., 2024), DAMRO (Gong et al., 2024), IMCCD (Li et al., 2025a), CATCH (Kan et al., 2024), IBD (Zhu et al., 2024), CAUSALMM (Zhou et al., 2024) and ICT (Chen et al., 2025). The results of CAI compared with more SOTA methods on MS-COCO POPE are shown in Table 8.

| Method | Random | | Popular | | Adversarial | | Average | |
|---|---|---|---|---|---|---|---|---|
| | Accuracy | F1-Score | Accuracy | F1-Score | Accuracy | F1-Score | Accuracy | F1-Score |
| Regular | 83.29 | 81.33 | 81.88 | 80.06 | 78.96 | 77.57 | 81.38 | 79.65 |
| ICD *(EMNLP'24 findings)* | 89.56 | 89.68 | 86.16 | 86.76 | 79.71 | 81.70 | 85.14 | 86.05 |
| OPERA *(CVPR'24)* | 89.20 | 88.81 | 86.64 | 86.62 | 81.24 | 81.38 | 85.70 | 85.60 |
| Woodpecker *(SCIS'24)* | 87.67 | 86.45 | 80.67 | 79.72 | 80.67 | 80.00 | 83.00 | 82.05 |
| M3ID *(CVPR'24)* | 86.20 | 84.51 | 84.77 | 83.17 | 82.53 | 81.14 | 84.50 | 82.94 |
| DAMRO *(EMNLP'24)* | 88.20 | 87.29 | 85.67 | 84.98 | 82.07 | 81.90 | 85.31 | 84.72 |
| IMCCD *(arXiv'25)* | 89.23 | 88.68 | 86.73 | 86.13 | 82.87 | 82.77 | 86.27 | 85.86 |
| CATCH *(ECCV'24)* | **90.43** | **90.13** | 87.07 | 86.56 | 83.17 | 83.18 | 86.89 | 86.62 |
| VDD *(arXiv'24)* | 90.00 | 88.79 | 85.91 | 84.40 | 83.52 | 82.20 | 86.48 | 85.13 |
| CAUSALMM *(ICLR'25)* | 88.93 | 88.10 | 87.13 | 87.26 | 83.70 | 82.78 | 86.59 | 86.05 |
| ICT *(CVPR'25)* | 90.11 | 90.03 | 87.50 | 87.60 | **84.43** | 83.74 | 87.35 | 87.12 |
| CAI(ours) | 89.87 | 89.43 | **88.32** | **87.95** | 84.27 | **84.41** | **87.49** | **87.22** |

Table 8: Result compared with more advanced methods on MS-COCO POPE.

To further demonstrate the superiority of CAI's performance, we additionally compare CAI with two advanced RL methods, including HADPO (Zhao et al., 2023) and HALVA (Sarkar et al., 2024). As shown in the Table 9 and Table 10, CAI achieves performance comparable to these RL methods and even surpasses them on discriminative tasks.

## E  DETAILED EXPERIMENTAL RESULTS OF MME

Detailed results on MME perception and cognition can be found in Table 11 and Table 12.

| Method | POPE | | | CHAIR ($\downarrow$) | | MME | | | |
|---|---|---|---|---|---|---|---|---|---|
| | Random | Popular | Adver. | CHAIR$_I$ | CHAIR$_S$ | Count | Exist. | Color | Posi. |
| LLaVA-1.5-7B | 83.29 | 81.88 | 78.96 | 15.9 | 52.8 | 124.67 | 175.67 | 151.00 | 114.00 |
| + HADPO | 86.00 | 85.10 | 82.90 | **11.0** | 38.2 | 133.30 | **190.00** | 158.30 | 136.70 |
| + HALVA | 86.40 | 85.50 | 83.20 | 11.7 | 41.4 | **165.00** | **190.00** | **175.00** | 135.00 |
| + CAI | **89.87** | **88.32** | **84.27** | 11.5 | **34.6** | 141.67 | **190.00** | 170.00 | **140.00** |

Table 9: Comparisons between CAI and RL works on POPE, CHAIR, and MME benchmarks.

| Method | HallusionBench | | | | | GAVIE | |
|---|---|---|---|---|---|---|---|
| | qAcc | fAcc | Easy aAcc | Hard aAcc | aAcc | Relevancy | Accuracy |
| LLaVA-1.5-7B | 10.55 | 20.86 | 41.67 | 29.77 | 46.04 | 8.20 | 6.42 |
| + HADPO | 11.21 | 19.08 | 42.86 | 39.19 | 47.46 | **8.84** | 6.30 |
| + HALVA | **13.85** | **21.48** | 42.71 | **40.81** | **47.95** | 8.72 | 6.46 |
| + CAI | 12.90 | 20.96 | **43.34** | 37.69 | 46.75 | 8.76 | **6.68** |

Table 10: Comparisons between CAI and RL works on HallusionBench and GAVIE benchmarks.

| Method | Artwork | Celebrity | Color | Count | Existence | Landmark | OCR | Position | Posters | Scene | Total |
|---|---|---|---|---|---|---|---|---|---|---|---|
| Regular | 102.20 | 113.59 | 151.00 | 124.67 | 175.67 | 129.95 | 92.00 | 114.00 | 127.82 | 148.30 | 1279.20 |
| VCD | 109.60 | 120.94 | 153.00 | 138.33 | 184.66 | 140.45 | 104.00 | 128.67 | 132.11 | 152.20 | 1363.96 |
| OPERA | **122.50** | 126.76 | 149.00 | 116.00 | 165.00 | 152.75 | **132.50** | 133.33 | 136.05 | 154.00 | 1387.89 |
| CAI(ours) | 120.25 | **135.88** | **170.00** | **141.67** | **190.00** | **158.50** | 120.00 | **140.00** | **140.48** | **157.00** | **1473.78** |

Table 11: Results on all MME perception-related tasks. The best performance of each is **bolded**.

| Method | Coding Reasoning | Commonsense Reasoning | Numerical Calculation | Text Translation | Total |
|---|---|---|---|---|---|
| Regular | 66.38 | 106.43 | 57.00 | 72.50 | 302.31 |
| VCD | 68.50 | 111.29 | 42.64 | 68.50 | 290.93 |
| OPERA | 62.50 | 119.29 | 37.50 | **82.50** | 301.79 |
| CAI(ours) | **75.00** | **122.86** | **57.50** | 50.00 | **305.36** |

Table 12: Results on all MME recognition-related tasks. The best performance is **bolded**.

# F  DOMAIN GENERALIZATION PERFORMANCE

In domain-specific tasks, the CAI method demonstrates certain generalization ability to some extent. Although caption queries are general instructions, they are extensively used during model pretraining. Activating the relevant attention patterns facilitates fine-grained visual information capture, thereby enhancing downstream task performance. To evaluate CAI's effectiveness in specific domains, we selected VQA-RAD (Lau et al., 2018) from the medical domain and the MMBench (Liu et al., 2024d) OCR subset. The experimental results of LLaVA-1.5-7b, as presented in the table 13, show consistent improvements over the baseline, indicating the CAI method's generalization ability.

| Domain | Dataset | Method | Accuracy |
|---|---|---|---|
| Medical | VQA-RAD | Greedy | 54.18% |
| | | CAI | 58.17% |
| OCR | MMBench | Greedy | 74.31% |
| | | CAI | 77.54% |

Table 13: Results on VQA-RAD and MMbench OCR subset.
.

## G  RESULTS ON MORE ADVANCED BENCHMARKS

The five commonly used hallucination evaluation benchmarks included in our paper follow the setups adopted in recent works. Using these benchmarks allows us to make fair and comprehensive comparisons with prior training-free methods. Furthermore, we additionally conduct experiments on more advanced hallucination evaluation benchmarks, including HallusionBench (Wu et al., 2024) and GAVIE (Liu et al., 2023a). As shown in Table 14, CAI also achieves improvements on these more critical evaluation.

| Method | HallusionBench | | | | | GAVIE | |
|---|---|---|---|---|---|---|---|
| | qAcc | fAcc | Easy aAcc | Hard aAcc | aAcc | Relevancy | Accuracy |
| LLaVA-1.5-7B | 10.55 | 20.86 | 41.67 | 29.77 | 46.04 | 8.20 | 6.42 |
| + CAI | **12.90** | **20.96** | **43.34** | **37.69** | **46.75** | **8.76** | **6.68** |
| Qwen-VL-Chat | 8.93 | 11.56 | 34.43 | 28.87 | 41.12 | 8.26 | 6.39 |
| + CAI | **11.47** | **13.57** | **35.60** | **31.87** | **43.93** | **8.63** | **6.60** |
| Qwen2.5-VL-7B | 16.43 | 31.01 | 59.73 | 34.93 | 50.79 | 9.20 | 8.09 |
| + CAI | **19.73** | **32.31** | **64.56** | **45.40** | **53.80** | **9.33** | **8.42** |

Table 14: Comparisons on HallusionBench and GAVIE benchmarks across different MLLMs.

## H  IMPACTS OF THE CLASSIFIER TYPES AND TRAINING DATA

### H.1  IMPACTS OF THE CLASSIFIER TYPES

Inspired by prior works (Li et al., 2023a; Bao et al., 2025; Zhang et al., 2024), which show that SVM effectively performs binary classification on high-dimensional internal model vectors, we adopt SVM as the classifier in our CAI framework. To further analyze the impacts of the classifier types, we implement **Logistic Regression (LR)** as an alternative classifier. The experimental results are shown in the table below. CAI with LR achieves performance nearly identical to CAI with SVM, as 95% of the Top-100 attention heads selected by both classifiers are the same. CAI with SVM exhibits a slight performance advantage, which aligns with findings in related work (Wang & Xu, 2025) and further confirms SVM's superior capability in classifying high-dimensional vectors.

| Model | POPE ($\uparrow$) | | | CHAIR ($\downarrow$) | | MME ($\uparrow$) | | | |
|---|---|---|---|---|---|---|---|---|---|
| | Random | Popular | Adversarial | $C_I$ | $C_S$ | Count | Exist. | Color | Posi. |
| LLaVA-1.5-7b | 83.29 | 81.88 | 78.96 | 15.9 | 52.8 | 124.67 | 175.67 | 151.00 | 114.00 |
| + CAI w/ SVM | **89.87** | **88.32** | **84.27** | **11.5** | **34.6** | **141.67** | **190.00** | **170.00** | **140.00** |
| + CAI w/ LR | 89.40 | 88.13 | 83.87 | 11.7 | 34.9 | 138.33 | **190.00** | **170.00** | 135.00 |

Table 15: Performance comparison between SVM and LR classifiers on POPE, CHAIR, and MME benchmarks. The best results are highlighted in **bold**.

### H.2  IMPACTS OF THE CLASSIFIER TRAINING DATA

To further investigate the amount of classifier training data, we randomly select distinct samples from the whole LLaVA-1.5-7B pre-training dataset (77K) and retrain the classifiers. We evaluate the classifying consistency of Top-$k$ heads using the **Overlap Ratio**, defined as $|H_n \cap H_{CAI}|/k$, where $n$ is the number of samples, $n \in \{100, 250, 500, 1500, 2000, 5000\}$; $k \in \{50, 100\}$; $H_n$ denotes heads identified by new classifiers and $H_{CAI}$ denotes heads identified in our primary results. The following table shows that the classifier's training is robust to data variations and amount, as the top-100 caption-sensitive attention heads which play a critical role in visual perception **predominantly coincide with** the CAI identified in the paper.

| Overlap Ratio | $n = 100$ | $n = 250$ | $n = 500$ | $n = 1000$ **(CAI)** | $n = 1500$ | $n = 2000$ | $n = 5000$ |
|---|---|---|---|---|---|---|---|
| $k = 10$ | 0.90 | 1.00 | 1.00 | 1.00 | 1.00 | 1.00 | 1.00 |
| $k = 50$ | 0.94 | 0.96 | 0.96 | 1.00 | 0.98 | 1.00 | 0.98 |
| $k = 100$ | 0.88 | 0.90 | 0.93 | 1.00 | 0.95 | 0.94 | 0.94 |

Table 16: Robustness analysis of classifier training. The high overlap ratios across varying sample sizes ($n$) and Top-$k$ attention heads demonstrate that the identified attention heads are consistent and robust to data amount variations compared to the primary setting ($n = 1000$).

## I   FINE-GRAINED ANALYSIS OF HYPERPARAMETERS

### I.1   GRID-SEARCH RESULTS OF LLaVA-1.5-7B ON POPE

In our experiments, we conducted a grid search to identify the optimal values of the hyperparameters $\alpha$ and $K$. We now provide the full grid-search results of LLaVA-1.5-7B on the POPE Adversarial benchmark, which gives a more clear and continuous view of how $\alpha$ and $K$ jointly affect CAI's performance. The best performance is achieved at $\alpha = 1.5$ and $K = 100$.

| Accuracy | $\alpha = 0$ | $\alpha = 0.5$ | $\alpha = 1.0$ | $\alpha = 1.25$ | $\alpha = 1.5$ | $\alpha = 1.75$ | $\alpha = 2.0$ |
|---|---|---|---|---|---|---|---|
| $K = 0$ | 78.96 | 78.96 | 78.96 | 78.96 | 78.96 | 78.96 | 78.96 |
| $K = 50$ | 78.96 | 79.31 | 79.86 | 80.18 | 80.50 | 80.32 | 80.40 |
| $K = 75$ | 78.96 | 79.82 | 80.13 | 80.44 | 80.77 | 80.59 | 80.31 |
| $K = 100$ | 78.96 | 81.07 | 82.50 | 83.47 | **84.27** | 84.14 | 84.00 |
| $K = 125$ | 78.96 | 80.79 | 82.16 | 83.28 | 84.10 | 83.82 | 83.51 |
| $K = 150$ | 78.96 | 80.24 | 81.47 | 82.53 | 83.20 | 82.97 | 82.68 |
| $K = 200$ | 78.96 | 79.91 | 81.18 | 82.12 | 83.00 | 82.76 | 82.43 |

Table 17: Grid-search results on POPE-Adversarial.

### I.2   GRID-SEARCH RESULTS OF LLaVA-1.5-7B ON CHAIR

CAI can achieve slightly better performance with task-specific hyperparameters in some generative tasks. As shown in the table, we conducted hyperparameter analysis on the CHAIR benchmark. The optimal parameters are found to be ($\alpha = 1.25$, $K = 125$ and performance = 34.3); nevertheless, the performance difference compared to the POPE-optimal parameters ($\alpha = 1.5$, $K = 100$ and performance = *34.6*) is minimal.

| $C_S \downarrow$ | $\alpha = 0$ | $\alpha = 1.0$ | $\alpha = 1.25$ | $\alpha = 1.5$ | $\alpha = 1.75$ | $\alpha = 2.0$ |
|---|---|---|---|---|---|---|
| $K = 0$ | 52.8 | 52.8 | 52.8 | 52.8 | 52.8 | 52.8 |
| $K = 50$ | 52.8 | 44.3 | 43.1 | 43.5 | 44.0 | 44.8 |
| $K = 75$ | 52.8 | 39.6 | 37.5 | 37.6 | 38.6 | 39.4 |
| $K = 100$ | 52.8 | 35.1 | 34.4 | *34.6* | 35.2 | 35.9 |
| $K = 125$ | 52.8 | 34.9 | **34.3** | 34.5 | 35.0 | 35.7 |
| $K = 150$ | 52.8 | 35.3 | 34.7 | 35.1 | 35.8 | 36.5 |
| $K = 200$ | 52.8 | 36.0 | 34.4 | 36.1 | 36.7 | 37.3 |

Table 18: Grid-search results on CHAIR.

Nevertheless, we observe that the optimal parameters identified on POPE Adversarial dataset **can generalize well to other discriminative and generative tasks** (e.g., MME, CHAIR, MMHal-Bench). This indicates that the fixed optimal hyperparameters can be effectively applied in real-world scenarios, demonstrating **CAI's ease of deployment and strong generalization capability**.

## J   DISCUSSION OF CAI ON THE FLY

CAI's shift vectors are precomputed for each model in our main experiments. This design is motivated by two key considerations:

(1) Robustness. As described in Section 4.1, each shift vector is obtained by averaging the attention differences over 1,000 diverse VQA samples. This averaging process aims to extract a general and robust direction for perceptual enhancement while diluting sample-specific semantic noise.

(2) Efficiency. Precomputation allows CAI to function as a plug-and-play module without introducing little additional inference-time cost.

To further explore the relationship between the extra computation required at inference time and the improvement achieved, we propose additional *on-the-fly* approach. Concretely, we dynamically compute each inference sample's attention difference between the "caption query" and the "non-caption query" and employ this sample-specific vector for intervention.

As shown in Table 19, the experimental results clearly demonstrate that:

(1) Slight performance drop. The *on-the-fly* variant remains effective, but consistently lower than the *precomputed* version. We believe this is because: the *precomputed* shift vector, which averages over 1,000 samples, yields a highly robust perception-enhancing direction. In contrast, the *on-the-fly* vector may inevitably carry more sample-dependent semantic noise, which limits its effectiveness.

(2) Substantial increase in inference cost. The *on-the-fly* approach requires two forward passes per sample, resulting in an 80% increase in inference latency.

In summary, the *precomputed* strategy adopted in our paper not only achieves better hallucination mitigation but also higher inference efficiency, making it a more practical choice for real-world applications.

| Method | Latency | POPE (↑) | | | CHAIR (↓) | |
|---|---|---|---|---|---|---|
| | | **Random** | **Popular** | **Adver.** | $C_I$ | $C_S$ |
| LLaVA-1.5-7B | 1.0× | 83.29 | 81.88 | 78.96 | 15.9 | 52.8 |
| + CAI (*precomputed*) | 1.0× | **89.87** | **88.32** | **84.27** | **11.5** | **34.6** |
| + CAI (*on the fly*) | 1.8× | 88.19 | 87.40 | 83.56 | 12.6 | 36.6 |

Table 19: Latency and performance comparisons between *precomputed* and *on-the-fly* approaches.

## K    DISCUSSION ON THE CAI INTERVENTION LAYERS

CAI method adds interventions across all model layers rather than targeting in a certain layer, based on prior studies on information flow (Li et al., 2025b; Golovanevsky et al., 2024; Neo et al., 2024; Meng et al., 2022), we argue that intervening on attention heads in a single layer alone cannot effectively enhance visual perception; these important attention heads must be activated or perturbed across layers to fully reinforce the visual information flow (Neo et al., 2024; Meng et al., 2022). Intervening only in shallow layers without affecting higher layers may impair perception, while intervening only in higher layers cannot fully strengthen the visual processing information flow (Li et al., 2025b), limiting CAI's ability to achieve optimal hallucination mitigation. As shown in Table 20, our experiments further confirm this: intervening on top-100 caption-sensitive heads in layers 0–10, 11–20 and 21-31 alone does not achieve optimal CAI performance and may even degrade model capability.

## L    DEEPER EVIDENCE OF CAI'S EFFECTIVENESS IN DISCRIMINATIVE SETTINGS

Previous works (Sarkar et al., 2024; Liu et al., 2023b) observed that the "yes-bias" in discriminative tasks arises because "models are finetuned on unbalanced datasets containing predominantly positive instructions" (Liu et al., 2023b), and thus represents the main form of LVLM's object hallucination. Furthermore, we computed the confusion matrices of LLaVA-1.5-7B on the POPE popular and random subsets. As shown in the Table, CAI substantially mitigates the "yes-bias," providing deeper evidence of CAI's effectiveness in discriminative settings.

| Method | POPE (↑) | | | CHAIR (↓) | |
|---|---|---|---|---|---|
| | Random | Popular | Adver. | $C_I$ | $C_S$ |
| LLaVA-1.5-7B | 83.29 | 81.88 | 78.96 | 15.9 | 52.8 |
| + CAI w/ 0-10 | 82.07 | 80.65 | 77.41 | 16.4 | 54.0 |
| + CAI w/ 11-20 | 87.16 | 85.83 | 82.52 | 13.0 | 38.2 |
| + CAI w/ 21-31 | 86.78 | 84.22 | 80.87 | 15.4 | 44.3 |
| + CAI (*Ours*) | **89.87** | **88.32** | **84.27** | **11.5** | **34.6** |

Table 20: **Ablation study on intervention layers.** We apply CAI to different blocks of layers to identify the most critical stages. The results show that intervening in the middle layers (11-20) yields more significant improvements than early or late layers, while the full CAI method achieves the best performance by coordinating across all identified heads.

| | Baseline | | CAI | |
|---|---|---|---|---|
| | **Pred: yes** | **Pred: no** | **Pred: yes** | **Pred: no** |
| **Golden: yes** | 1360 | 140 | 1277 | 223 |
| **Golden: no** | 274 | 1226 | 120 | 1380 |

Table 21: **Confusion matrix on POPE-Popular.** Compared to the baseline, CAI significantly reduces the number of "No" samples incorrectly predicted as "Yes" (from 274 to 120), effectively mitigating the "Yes Bias".

| | Baseline | | CAI | |
|---|---|---|---|---|
| | **Pred: yes** | **Pred: no** | **Pred: yes** | **Pred: no** |
| **Golden: yes** | 1340 | 160 | 1290 | 210 |
| **Golden: no** | 197 | 1303 | 83 | 1417 |

Table 22: **Confusion matrix on POPE-Random.** Similarly, on the random split, CAI drops the false positive rate drastically (from 197 to 83), effectively mitigating the "Yes Bias".

## M  CASE STUDY FOR CAPTION QUERIES

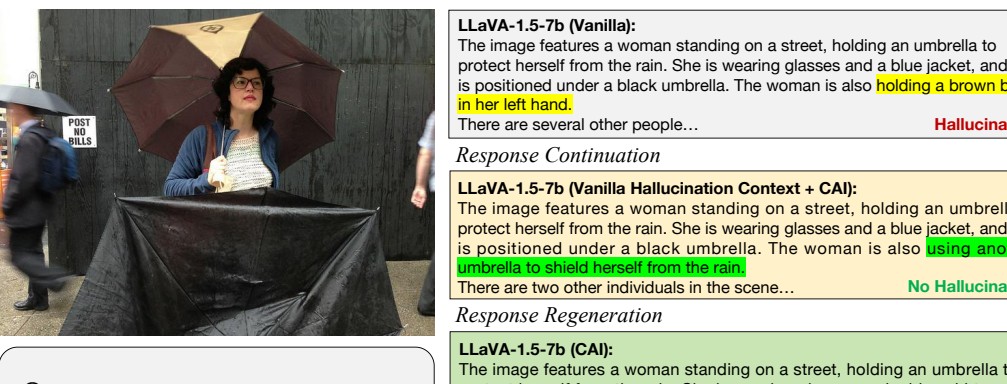

Figure 7: Case study of caption task on CHAIR.

CAI remains effective in caption task, which is attributed to the enhancement in visual attention. As shown in Figure 7, CAI effectively mitigates object hallucination not only during the regeneration of new responses, but also when extending hallucinated contexts, highlighting its fine-grained, token-level object hallucination mitigation capability.

# N  CASE STUDIES FOR NON-CAPTION QUERIES

More case studies when fed non-caption queries are shown as follows.

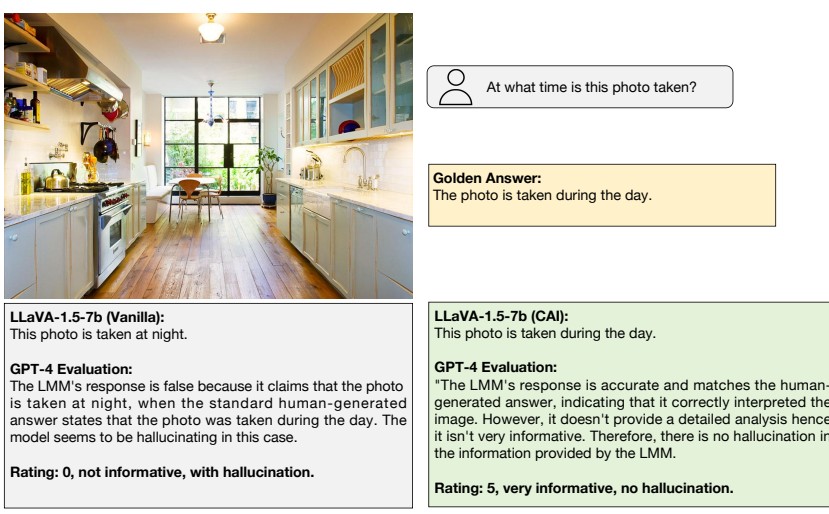

Figure 8: Non-caption query case of LLaVA-1.5-7b on MMHal-Bench.

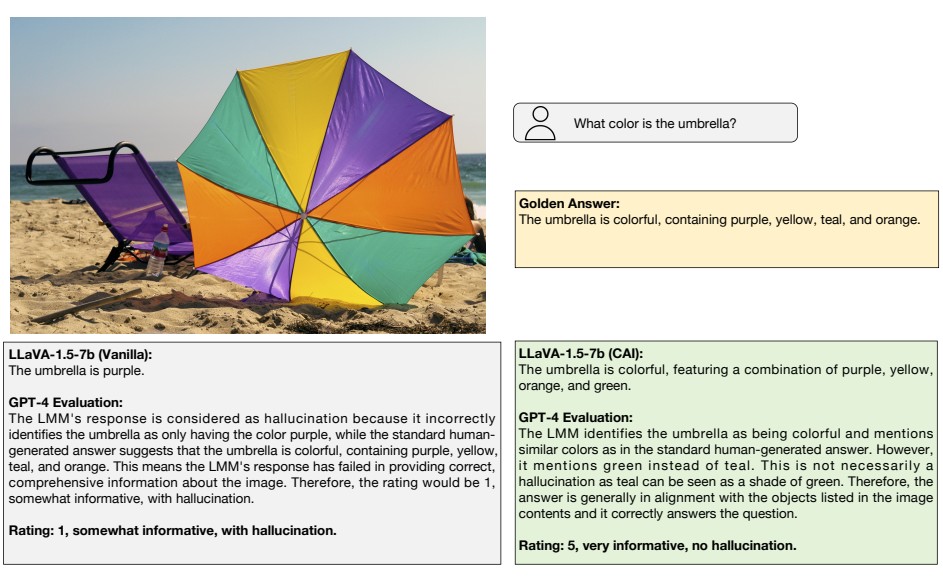

Figure 9: Non-caption query case of LLaVA-1.5-7b on MMHal-Bench.

# O  USAGE OF LARGE LANGUAGE MODELS

## O.1  ASSISTANCE FOR WRITING POLISHING

During the writing process, we employed GPT-4o (Hurst et al., 2024) for writing polishing. In particular, we utilized LLM assistance in the method section to articulate more clearly the motivation, implementation, and corresponding mathematical formulations of the CAI approach. In addition, we applied moderate polishing to the abstract and introduction to further enhance the readability and academic rigor of the paper.

### O.2 Assistance for Benchmark Evaluation

In conducting experiments with MMHal-Bench, we employed GPT-4 (Achiam et al., 2023) as an evaluation tool to assess hallucination capabilities.

