# OpenReview forum: "CAI: Caption-Sensitive Attention Intervention for Mitigating Object Hallucination in Large Vision-Language Models"
_ICLR.cc/2026/Conference — Submitted to ICLR 2026_

### Official Review · Reviewer_bUqS · 2025-10-27

**Soundness:** 3
**Presentation:** 3
**Contribution:** 3
**Rating:** 6
**Confidence:** 3

**Summary:**

This paper introduces CAI (Caption-Sensitive Attention Intervention), a training-free, plug-and-play method to reduce object hallucination in Large Vision-Language Models (LVLMs). CAI identifies caption-sensitive attention heads—those with stronger visual activation under caption prompts—and applies inference-time attention shift interventions to enhance visual grounding. Experiments on five LVLMs and benchmarks show up to 6.03% hallucination reduction, surpassing prior training-free methods like VCD, OPERA, and VTI with little inference overhead.

**Strengths:**

- The discovery that caption prompts systematically enhance visual attention, and exploiting this through attention-head steering, is novel and insightful.
- Experiments are robust, covering 5 models × 5 benchmarks, with detailed ablations, latency analysis, and query optimization strategies.
- The method is described step-by-step and is easy to follow.
- Offers an effective alternative to decoding-based approaches, with broad implications for multimodal interpretability and reliability research.

**Weaknesses:**

The intervention directly adds attention-based perturbations to hidden states (Eq. 9). Without formal constraints, it remains unclear how these modified representations remain within the model’s valid latent manifold. Could you elaborate on how to guarantee that the modified attention remains within the LVLM’s representational manifold? Why does this intervention not introduce unintended artifacts?

Could you clarify how the intervention is applied to the network (does it apply to all layers or only specific ones)? Have the authors analyzed inter-layer dependencies—for example, whether interventions at lower layers amplify or dampen representations at higher layers? From such analysis, is it possible to determine the layers where interventions would be most effective?

**Questions:**

See weaknesses

---

> ### Author Response · Authors · 2025-11-19
>
> We thank the reviewer for these profound insights. Next, we will explain your concerns point by point:
> ### **1. Response to the concern on valid manifold.**
> We believe that CAI **guides latent transition** between two valid manifold, specifically from non-caption latent manifold to caption latent manifold.
>
> - **From the mathematical theoretical perspective:** Our intervention vector (defined in Equation 8) is derived from the average difference between two valid manifold. As the essence of CAI intervention is an **affine translation**, the post-intervention manifold lies within the subspace associated with caption queries. Concretely, we are not pushing hidden states off the “valid manifold”; instead, we are moving them **along a “fine-grained perception direction” axis** on the manifold, leading to a state of sufficient visual attention.
>
> - **From the empirical perspective:** CAI does not produce unintended artifacts affecting general task performance: (1) CAI improves general performance on MME benchmark and enhances the informativeness of generated text on MMHal-Bench. (2) Experiments in Table 5 shows that even doubling the intervention magnitude does not harm text quality. These empirical results demonstrate that CAI interventions are safe and controllable, the post-intervention manifold remains within the LVLM’s representational manifold and does not include harmful artifacts.
>
> ### **2. Discussion on the CAI intervention layers.**
> (1) CAI method adds interventions **across all model layers rather than targeting in a certain layer**. We identify the most important heads across layers using the “caption-sensitive attention heads probing” technique described in Section 3.2.
>
> (2) We had identified the critical layers for visual perception, through two complementary approaches in the paper:
>
> - Results from Section 2’s quantitative analysis demonstrate that caption-induced visual attention enhancements primarily occur in mid layers.
> - Results from the probing classifier in Figure 5 shows that the caption-sensitive heads are mainly located between layers 7 and 20.
>
>     Thus, as stated in line 393, both quantitative analyses and probing results consistently indicate that mid layers (approximately layers 7–20) are the most effective layers for fine-grained visual perception.
>
> (3) Additionally, based on prior studies on information flow [1,2,3,4], we argue that intervening on attention heads in a single layer alone cannot effectively enhance visual perception; these **important attention heads must be activated or perturbed across layers to fully reinforce the visual information flow**[3,4]. Intervening only in shallow layers without affecting higher layers may impair perception, while intervening only in higher layers cannot fully strengthen the visual processing information flow [1], limiting CAI’s ability to achieve optimal hallucination mitigation. Our experiments further confirm this: intervening on top-100 caption-sensitive heads in layers 0–10, 11–20 and 21-31 alone does not achieve optimal CAI performance and may even degrade model capability.
>
> |               | POPE ran. | POPE pop. | POPE adver. | CHAIRi$\downarrow$ | CHAIRs$\downarrow$ |
> | ------------- | :-------: | :-------: | :---------: | :----------------: | :----------------: |
> | LLaVA-1.5-7B  |   83.29   |   81.88   |    78.96    |        15.9        |        52.8        |
> | +CAI w/ 0-10  |   82.07   |   80.65   |    77.41    |        16.4        |        54.0        |
> | +CAI w/ 11-20 |   87.16   |   85.83   |    82.52    |        13.0        |        38.2        |
> | +CAI w/ 21-31 |   86.78   |   84.22   |    80.87    |        15.4        |        44.3        |
> | +CAI          | **89.87** | **88.32** |  **84.27**  |      **11.5**      |      **34.6**      |
>
> **References**
>
> [1] Causal Tracing of Object Representations in Large Vision Language Models: Mechanistic Interpretability and Hallucination Mitigation (AAAI2026 Oral)
>
> [2] What Do VLMs NOTICE? A Mechanistic Interpretability Pipeline for Gaussian-Noise-free Text-Image Corruption and Evaluation (NAACL2025)
>
> [3] Towards Interpreting Visual Information Processing in Vision-Language Models (ICLR2025)
>
> [4] Locating and Editing Factual Associations in GPT (NeurIPS2022)
>
> ---
> ### _**We sincerely thank for the reviewer‘s careful reading and insightful feedback. We hope these clarifications could address your concerns.**_

---

> > ### Comment · Reviewer_bUqS · 2025-11-27
> >
> > Thank the authors for their follow-up and feedback. Most of my concerns are addressed, and I maintain my positive rating.

---

> > > ### Author Response · Authors · 2025-11-28
> > >
> > > Dear Reviewer bUqS,
> > >
> > > We sincerely appreciate your recognition of our rebuttal and maintain the positive rating. We truly value the time you have spent on our work and will ensure that the final manuscript is further polished based on your feedback.
> > >
> > > Thank you once again!
> > >
> > > Best regards,
> > >
> > > Authors

---

### Official Review · Reviewer_mtt9 · 2025-10-27

**Soundness:** 1
**Presentation:** 3
**Contribution:** 1
**Rating:** 2
**Confidence:** 5

**Summary:**

This paper introduces a training-free approach to reduce object hallucinations in LVLMs, especially in non-caption tasks. The authors claim that during non-caption VQAs, there is a lack of visual attention. To address this, they precompute a shift score in the attention heads using a set of images and caption queries, which is then used at inference time to adjust the attention scores for a given input image-question pair. This idea is implemented on three LVLMs and evaluated on five hallucination benchmarks.

**Strengths:**

* The paper is well-written and easy to follow.
* This paper addresses object hallucination, which remains a critical problem in LVLMs.
* The idea of adjusting visual attention for queries where the models originally fail to provide strong visual focus seems interesting.
* Addressing hallucination through intervention at inference is an efficient choice, compared to techniques that require heavy training.

**Weaknesses:**

* In Fig. 1: "Are there both a helmet and a motorcycle in the image?" The reason LLaVA 1.5 answered "Yes" is because (most likely) the model heavily suffered from a "yes" bias. This issue has been highlighted in prior works, e.g., *Mitigating Object Hallucination in LVLMs via Data-augmented Phrase-level Alignment, ICLR 2025*; you can do a quick study, check the confusion matrix with and without CAI -- you will most likely see that the improvements mainly stem from YES to NO.

* The experimental setup is very weak; the proposed method is applied on old models—LLaVA 1.5 (released in 10/2023), Qwen VL Chat (released in 08/2023), and LLaVA Next (released in 01/2024). Current models are much better than the 2023 models. If the proposed method can improve the performance of current LVLMs (e.g., QwenVL2.5/3, InternVL2.5/3, LLaVA-One-Vision1/1.5), it would be of interest to the community.

* The baselines used in comparisons are old; both OPERA and VCD were released in 2023. You should compare against more recent decoding methods. You should also include training-based methods in the comparisons, such as HADPO and HALVA, among other offline RL approaches.

* The benchmarks utilized in this work, such as POPE and CHAIR, are old and do not accurately evaluate the hallucination of LVLMs. POPE evaluates object existence based on 500 images and does not cover other forms of object hallucination, such as object attributes and relations. Similarly, CHAIR also evaluates only 500 images from MSCOCO using limited ground-truth information. Additionally, MMHal-Bench has only 96 images to test on. While these benchmarks have been commonly used in the past, they lack diversity and rigor, and there is a need for more critical evaluation. The authors can explore better benchmarks like HallusionBench, AMBER, M-HalDetect, and GAVIE.

* l.270: Are you using a precomputed shift vector for each model, or are you computing this shift vector on the fly? If it is computed on the fly, I would be interested in understanding the relationship between the extra computation required at inference time and the improvement achieved.

* LLaVA-Next is not included when evaluated on CHAIR, MMHal-Bench, and MHumanEval!

* In l.324, the authors claimed to achieve SOTA performance without comprehensive comparisons.

* The authors mentioned about using 5 LVLMs in the abstract but they mention about 3 in the paper!

**Questions:**

Please see the weaknesses above.

---

> ### Author Response · Authors · 2025-11-19
>
> **We sincerely appreciate your constructive and insightful feedback. Next, we will explain your concerns point by point:**
> ### **1. Quick study of confusion matrix on POPE benchmarks.**
> Thank you for providing this insightful work [1]. After carefully study the related works [2,3], we agree that the “yes-bias” observed in discriminative tasks arises because *"models are finetuned on unbalanced datasets containing predominantly positive instructions"* [2], and thus represents a form of suffering from hallucination.
>
> Following your suggestion, we computed the confusion matrices of LLaVA-1.5-7B on the POPE _popular_ and _random_ subsets. As shown in the table, and consistent with your expectation, CAI substantially mitigates the “yes-bias,” providing deeper evidence of CAI’s effectiveness in discriminative settings. We will cite the referenced work and include these results in the appendix.
>
> Moreover, CAI remains effective in more complex and challenging **generative tasks**. CAI achieves consistent improvements across mainstream benchmarks( CHAIR, MMHal-Bench MHumanEval) and domain-specific benchmarks (VQA-RAD and MMBench), demonstrating that CAI effectively enhances the model’s fine-grained visual perception rather than only mitigating "Yes-bias".
>
> ***Confusion matrix on POPE popular***
>
> | Baseline        | Pred: yes | Pred: no | CAI             | Pred: yes | Pred: no |
> | :-------------- | :-------- | :------- | :-------------- | :-------- | :------- |
> | **Golden: yes** | 1360      | 140      | **Golden: yes** | 1277      | 223      |
> | **Golden: no**  | 274       | 1226     | **Golden: no**  | 120       | 1380     |
>
> ***Confusion matrix on POPE random***
>
> | Baseline        | Pred: yes | Pred: no | CAI             | Pred: yes | Pred: no |
> | :-------------- | :-------- | :------- | :-------------- | :-------- | :------- |
> | **Golden: yes** | 1340      | 160      | **Golden: yes** | 1290      | 210      |
> | **Golden: no**  | 197       | 1303     | **Golden: no**  | 83        | 1417     |
>
> [1] Mitigating Object Hallucination in MLLMs via Data-Augmented Phrase-Level Alignment (ICLR2025)
>
> [2] Mitigating Hallucination in Large Multimodal Models via Robust Instruction Tuning (ICLR2024)
>
> [3] Evaluating Object Hallucination in Large Vision-Language Models (EMNLP2023)
>
> ### **2. Response to the suggestion on baseline models.**
> We sincerely appreciate your suggestion regarding baseline models. We would first like to clarify that our experiments are not limited to older models and well-designed:
>
> (1) **"Older" models for fair comparisons.** Mainstream prior work provides open-source implementations and experimental setups only for "older" models such as LLaVA-1.5-7B and Qwen-VL-Chat.
>
> (2) **More advanced models already provided in Appendix C.**  We had already included experimental results on more advanced LVLMs (Qwen2-VL-7B and InternVL2-8B) to show that CAI remains strong hallucination mitigation capability (as noted in line 253 of the main text, Appendix C).
>
> (3) **Added advanced model.** To further address your concern, we have additionally use the model you suggested (Qwen2.5-VL-7B) as a new baseline. As shown in the table, CAI also exhibits powerful hallucination mitigating performance.
>
> |               | POPE ran. | POPE pop. | POPE adver. | CHAIRi$\downarrow$ | CHAIRs$\downarrow$ |  MME(count\|exist.\|color\|posi.)   |  MME   |
> | ------------- | :---------: | :----------: | :--------------: | :----------------: | :----------------: | :--------------------------------------------: | :---------: |
> | Qwen2.5-VL-7B |    88.20    |    87.47     |      86.37       |        8.7         |        37.2        |     153.33\|**195.00**\|**190.00**\|145.00     |   2160.02   |
> | +CAI          |  **90.70**  |  **88.87**   |    **87.17**     |      **8.0**       |      **32.6**      | **160.00**\|**195.00**\|**190.00**\|**150.00** | **2233.86** |
>
> |               | HallusionBench(qAcc\|fAcc\|Easy aAcc\|Hard aAcc\|aAcc) | GAVIE(Relevancy\|Accuracy) |
> | ------------- | :----------------------------------------------------: | :------------------------: |
> | Qwen2.5-VL-7B |           16.43\|31.01\|59.73\|34.93\|50.79            |         9.20\|8.09         |
> | +CAI          |         **19.73\|32.31\|64.56\|45.40\|53.80**          |       **9.33\|8.42**       |

---

> ### Author Response · Authors · 2025-11-19
>
> ### **3. Response to the suggestion on baseline methods.**
>
> Thank you very much for raising concerns regarding the baseline methods.
>
> **Following prior work, our paper already compares CAI against 13 training-free methods on POPE (as noted at line-254 and Appendix D).** These include:
>
> (1) Two classical early methods: OPERA(2023) and VCD(2023);
>
> (2) Recent relevant work: VTI(2024) and PAI(2024);
>
> (3) A broad set of additional training-free methods reported in Appendix D, including ICD(2024), Woodpecker(2023), M3ID(2024), DAMRO(2024), IMCCD(2025), CATCH(2024), VDD(2024), CAUSALMM(2025), and ICT(2025).
>
> These baselines cover a wide range of decoding methods, and we believe this comprehensive comparisons sufficiently demonstrate the performance advantages of CAI.
>
> Moreover, to further address your concern (although CAI is training-free and thus not directly comparable to RL–based approaches), we additionally include results comparing CAI with the two RL methods you mentioned (**HADPO and HALVA**). As shown in the table, CAI achieves performance comparable to these RL methods and even surpasses them on discriminative tasks. This further validates the strong hallucination mitigation capability of CAI.
>
> |              | POPE ran. | POPE pop. | POPE adver. | CHAIRi$\downarrow$ | CHAIRs$\downarrow$ |      MME(count\|exist.\|color\|posi.)      |
> | :----------- | :---------: | :----------: | :--------------: | :----------------: | :----------------: | :----------------------------------------------: |
> | LLaVA-1.5-7B |    83.29    |    81.88     |      78.96       |        15.9        |        52.8        |       124.67\|175.67\|151.00\|114.00       |
> | +HADPO       |    86.00    |    85.10     |      82.90       |      **11.0**      |        38.2        |        133.30\|**190.00**\|158.30\|136.70        |
> | +HALVA       |    86.40    |    85.50     |      83.20       |        11.7        |        41.4        | **165.00**\|**190.00**\|**175.00**\|135.00 |
> | +CAI         |  **89.87**  |  **88.32**   |    **84.27**     |        11.5        |      **34.6**      |   141.67\|**190.00**\|170.00\|**140.00**   |
>
> |              | HallusionBench(qAcc\|fAcc\|Easy aAcc\|Hard aAcc\|aAcc) | GAVIE(Relevancy\|Accuracy) |
> | ------------ | :----------------------------------------------------: | :------------------------: |
> | LLaVA-1.5-7B |           10.55\|20.86\|41.67\|29.77\|46.04            |         8.20\|6.42         |
> | +HADPO       |           11.21\|19.08\|42.86\|39.19\|47.46            |       **8.84**\|6.30       |
> | +HALVA       |   **13.85**\|**21.48**\|42.71\|**40.81**\|**47.95**    |         8.72\|6.46         |
> | +CAI         |         12.90\|20.96\|**43.34**\|37.69\|46.75          |       8.76\|**6.68**       |
>
> ### **4. Response to the suggestion on benchmarks selection.**
> Thank you very much for your constructive suggestions regarding the benchmarks selection.
>
> - First, the five commonly used hallucination evaluation benchmarks included in our paper follow the setups adopted in recent works. **Using these benchmarks allows us to make fair and comprehensive comparisons with prior training-free methods.**
>
> - Second, we additionally conducted experiments on the benchmarks you recommended—**HallusionBench** and **GAVIE**. As shown in the table, CAI also achieves improvements on these more critical evaluation.
>
> |               | HallusionBench(qAcc\|fAcc\|Easy aAcc\|Hard aAcc\|aAcc) | GAVIE(Relevancy\|Accuracy) |
> | ------------- | :----------------------------------------------------: | :------------------------: |
> | LLaVA-1.5-7B  |           10.55\|20.86\|41.67\|29.77\|46.04            |         8.20\|6.42         |
> | +CAI          |         **12.90\|20.96\|43.34\|37.69\|46.75**          |       **8.76\|6.68**       |
> | Qwen-VL-Chat  |            8.93\|11.56\|34.43\|28.87\|41.12            |         8.26\|6.39         |
> | +CAI          |         **11.47\|13.57\|35.60\|31.87\|43.93**          |       **8.63\|6.60**       |
> | Qwen2.5-VL-7B |           16.43\|31.01\|59.73\|34.93\|50.79            |         9.20\|8.09         |
> | +CAI          |         **19.73\|32.31\|64.56\|45.40\|53.80**          |       **9.33\|8.42**       |

---

> ### Author Response · Authors · 2025-11-19
>
> ### **5. Discussion of CAI on the fly.**
> Thank you for your thoughtful discussion regarding the implementation mechanism of CAI’s shift vector.
>
> First, in our experiments, shift vectors are ***precomputed* for each model**. This design is motivated by two key considerations:
>
> (1) **Robustness.** As described in Section 4.1 and Appendix B.2.2, each shift vector is obtained by averaging the attention differences over 1,000 diverse VQA samples. This averaging process aims to extract **a general and robust direction for perceptual enhancement** while diluting sample-specific semantic noise.
>
> (2) **Efficiency.** Precomputation allows CAI to function as a plug-and-play module without introducing little additional inference-time cost (as shown in Table 6).
>
> Second, we believe your proposed **_on-the-fly_ strategy** is both insightful and interesting. Concretely, we dynamically compute each inference sample's attention difference between the “caption query” and the “non-caption query” and employ this sample-specific vector for intervention. The results on LLaVA-1.5-7B are shown in the table below:
>
> |                      | Latency | POPE ran. | POPE pop. | POPE adver. | CHAIRi$\downarrow$ | CHAIRs$\downarrow$ |
> | :------------------- | :-----: | :---------: | :----------: | :--------------: | :----------------: | :----------------: |
> | LLaVA-1.5-7B         |   1.0$\times$  |    83.29    |    81.88     |      78.96       |        15.9        |        52.8        |
> | +CAI (*precomputed*) |   1.0$\times$  |  **89.87**  |  **88.32**   |    **84.27**     |      **11.5**      |      **34.6**      |
> | +CAI (*on the fly*)  |   1.8$\times$  |    88.19    |    87.40     |      83.56       |        12.6        |        36.6        |
>
> The experimental results clearly demonstrate that:
>
> (1) **Slight performance drop.** The _on-the-fly_ variant remains effective, but consistently lower than the _precomputed_ version. We believe this is because: the _precomputed_ shift vector, which averages over 1,000 samples, yields a highly generalized and robust perception-enhancing direction. In contrast, the _on-the-fly_ vector may inevitably carry more sample-dependent semantic noise, which limits its effectiveness.
>
> (2) **Substantial increase in inference cost.** The _on-the-fly_ approach requires two forward passes per sample, resulting in an **80% increase in inference latency**.
>
> In summary, the *precomputed* strategy adopted in our paper not only achieves better hallucination mitigation but also higher inference efficiency, making it a more practical choice for real-world applications.
>
> ### **6. Additional results of LLaVA-Next.**
> We provide additional results of LLaVA-Next on CHAIR, MMHal-Bench, and MHumanEval.
>
> |            | CHAIRi$\downarrow$  |  CHAIRs$\downarrow$  | MME(color\|exist.\|count\|posi.) | MMHal-Bench(Score$\uparrow$\|VH$\downarrow$) | MHumanEvalVH$\downarrow$ |
> | ---------- | :-----: | :------: | :--------------------------------------: | :-----------------------------------------: | :----------------------: |
> | LLaVA-Next |  10.5   |   40.0   |      151.67\|180.00\|105.00\|150.00      |                 2.57\|55.8                  |           65.4           |
> | +CAI       | **8.9** | **33.3** |    **177.50\|190.00\|135.00\|155.00**    |               **3.12\|48.9**                |         **61.0**         |
>
> ### **7. Response to the concern on comprehensive comparisons.**
> Thank you for raising concerns regarding the performance of CAI. Based on the extensive experiments reported in our submission as well as the additional results provided in the rebuttal, we have compared CAI against a total of **15 hallucination-mitigation methods**—including **13 training-free approaches** and **2 RL–based methods**. We believe these comprehensive comparisons sufficiently demonstrate that CAI mostly achieves excellent performance.
>
> ### **8. Clarification about the numbers of reported LVLMs.**
> We sincerely remind the reviewer that we had reported experimental results on three LVLMs in the main text, and results on another two advanced LVLMs (**as stated in line 253, Appendix C**). Therefore, we believe the description in the abstract is correct.
>
> ---
> ### _**We sincerely thank for the reviewer‘s careful reading and insightful feedback. We hope these clarifications could address your concerns, and we would be very grateful if the reviewer could consider a higher rating.**_

---

> ### Author Response · Authors · 2025-11-24
>
> Dear Reviewer mtt9,
>
> Thank you very much for your constructive and thoughtful feedback. We kindly request that you confirm whether our responses have addressed your questions. If there are any additional questions remaining, please do not hesitate to let us know!
>
> Additionally, if our rebuttal has addressed your comments, we would greatly appreciate your consideration in updating your rating accordingly.
>
> We sincerely value your time and consideration and look forward to your feedback.
>
> Best regards,
>
> Authors

---

> > ### Comment · Reviewer_mtt9 · 2025-11-25
> >
> > Dear Authors,
> >
> > Thanks for providing the detailed response to my comments and conducting those additional experiments.
> > I will increase my score.
> >
> > All the best.

---

> > > ### Author Response · Authors · 2025-11-25
> > >
> > > Dear Reviewer mtt9,
> > >
> > > We sincerely appreciate your recognition of our rebuttal and the increased score. We truly value the time you have spent on our work and will ensure that the final manuscript is further polished based on your feedback.
> > >
> > > Thank you once again!
> > >
> > > Best regards,
> > >
> > > Authors

---

### Official Review · Reviewer_bzdp · 2025-10-31

**Soundness:** 3
**Presentation:** 3
**Contribution:** 3
**Rating:** 6
**Confidence:** 4

**Summary:**

The paper introduces a method for mitigating the visual hallucinations in LVLMs, whose main idea is to modify the attention heads responsive to visual tokens during inference time. These attention heads are selected from a subset of heads which are most activated when fed caption queries.

More specifically, the method consists of two steps:
- [offline] Selection of attention heads to modify - the authors propose to identify these by training binary classifiers on last token’s modified attention scores when answering a caption query and a non-caption query for a VQA task. Among the $l \times h$ classifiers for each layer and each head, the top $K$ are selected for modification.
- [offline] Estimation of the correction (shift) vector - calculated as the difference between the original output attention scores of a caption and a non-caption query for a VQA task.

At inference time, the attention scores of the identified $K$ heads are modified with the calculated shift vector.

**Strengths:**

- Well written method section.
- Improvements across models and datasets.

**Weaknesses:**

My concern regarding the presented method is its practicality - the method works well for object recognition but might degenerate some other tasks than require textual understanding (like text translation in Figure 4). I suppose this is due to the design of the process for selecting the intervention heads, which masks the attention towards textual tokens when calculating the modified attention output. This implies that the binary classifiers are trained on synthetic representations and selecting best heads based on their accuracy might lead to poor results. For example, if a head achieves high accuracy on synthetic representations but is also critical for a textual task (like translation), CAI will hurt the performance of the LVLM.

**Questions:**

- Judging from tables 10 and 11, it seems that CAI improves OCR but degenerates translation capabilities which rely on text. How does one need to modify your approach to fix this? Would the translation performance improve solely by tuning the hyperparameters or does one need to invoke the correct heads - eg, by training the classifiers on specific data?
- I don’t understand how the optimal query was determined. Which metrics did you look at for the selection? It is also not clear which query setting was used in the experiments - the optimal query or the ensemble query?
- What is the role of the data used to train the classifiers? Are these models robust to the choices and amount of the data samples?
- I do not see which experiment shows that it is necessary to keep both $\alpha$ and $K$ as hyperparameters. What kind of results would you get when only tuning $K$ for a constant $\alpha = 1$?


Nits:
- Figure 5 and 6 are lacking details. For FIgure 6 - What is K on the left, what is alpha on the right? On which dataset and model were these numbers obtained? What is Percentage on y-axis? Do you mean accuracy? For Figure 5 - from which experiment were these results taken fro?
- In line 197, should it be $b$-th rather than  $b^′$-th?

---

> ### Author Response · Authors · 2025-11-19
>
> We thank the reviewer for the constructive and insightful feedback. Next, we will explain your concerns point by point:
> ### **1. Response to the concerns of practicality on textual tasks (Weakness&Question1).**
> Thank you for raising concerns regarding the practicality of CAI and its influence on text-related tasks. Your analysis is highly insightful and inspiring.
> - **Regarding the method design:** Masking textual tokens during probing is an _intentional_ design aimed at precisely identifying attention heads that are critical to **visual perception** rather than **textual semantics**. By accurately locating and intervening on these heads, CAI **achieves consistent improvements on both hallucination-mitigation and most of general tasks**, thereby effectively addressing the core challenge posed in our work.
> - **Regarding the performance on textual tasks**, we agree with your concern that certain attention heads may be critical to both visual and textual tasks. To eliminate attention heads that are jointly sensitive to both visual and textual signals, we add a parallel **Textual Probe Technique** to identify text-sensitive heads. Concretely, we **mask visual tokens instead of textual tokens during probing** and identify a set of text-sensitive heads. We then intervene exclusively on the top-100 vision-sensitive attention heads but do not rank among the top-100 text-sensitive heads.
>
> |                      | POPE ran. | POPE pop. | POPE adver. | CHAIRi$\downarrow$ | CHAIRs$\downarrow$ | MME text translation |
> | -------------------- | :-------: | :-------: | :---------: | :----------------: | :----------------: | :------------------: |
> | LLaVA-1.5-7b         |   83.29   |   81.88   |    78.96    |        15.9        |        52.8        |        72.50         |
> | + CAI                | **89.87** | **88.32** |  **84.27**  |      **11.5**      |      **34.6**      |        50.00         |
> | + CAI w/ vision-only |   88.76   |   87.40   |    83.44    |        13.1        |        36.5        |      **75.00**       |
>
> As shown in the table, this modification successfully resolves the performance drop in text translation, albeit with a slight reduction in hallucination-mitigation performance compared to the original CAI. **This trade-off is likely unavoidable, as the visual and textual capabilities of LVLMs may not be fully disentangled.** We will investigate this in detail in future work.
>
> ### **2. Response to the selection of the optimal query.**
> Thank you for your question regarding the _optimal query_. We would like to offer additional clarification on the details presented in the submission:
> - **Selection metrics of the optimal query:** As stated in lines 373–376, we have clearly defined the concept of _optimized queries_:  _“Caption queries with minimal necessary shift cost yield better hallucination mitigation performance, and we term these queries as optimized queries.”_  The shift cost, as described in Section 2 and Equation 12 in Appendix A, represents the sum of attention-weight differences between answering caption queries and non-caption queries on a dataset.
> - **Query setting used in the main experiments:** All results reported in the main experiments were obtained using the _optimal query_. Appendix B (lines 779–795) provides detailed captions for all candidate caption queries as well as the optimal query selected for each model. Furthermore, the ensemble method was used only in our analysis (Section 5.1).

---

> ### Author Response · Authors · 2025-11-19
>
> ### **3. Response to the question of classifier training data.**
> Thank you for your question regarding the classifier training data. We would like to response from the following perspectives:
> - **Role of the classification data:** As stated in line 269, _“we utilize 1000 task-diverse VQAs from the LLaVA-1.5-7B pretraining dataset, each paired with a specific caption query.”_  These VQAs are general and cover a wide range of real-world scenarios; the questions themselves directly serve as _non-caption queries_. The corresponding caption queries are selected from the caption query candidates listed in Appendix B.2.1.
> - **Robustness of the classification data:** We used the same classification dataset **across five different LVLMs**, and all models exhibited consistent improvements in hallucination mitigation **across five widely used benchmarks**. Furthermore, as described in Appendix F, this dataset demonstrates strong robustness **even for other _domain-specific_ benchmarks** (VQA-RAD, MMBench OCR).
> - **Amount of classification data:** To further investigate this, we randomly select distinct samples from the whole LLaVA-1.5-7B pre-training dataset (77K) and retrain the classifiers. We evaluate the classifying consistency of Top-$k$ heads using the **Overlap Ratio**, defined as $|H_{n} \cap H_{CAI}|/k$, where $n$ is the number of samples, $n \in (100,250,500,1500,2000,5000)$; $k\in(50,100)$; $H_{n}$ denotes heads identified by new classifiers and $H_{CAI}$ denotes heads identified in our primary results. The following table shows that the classifier's training is robust to data variations and amount, as the top-100 caption-sensitive attention heads which play a critical role in visual perception **predominantly coincide with** the CAI identified in the paper.
>
> | Overlap Ratio | $n=100$ | $n=250$ | $n=500$ | $n=1000$(CAI) | $n=1500$ | $n=2000$ | $n=5000$ |
> | :-----------: | :-----: | :-----: | :-----: | :-----------: | :------: | :------: | :------: |
> |    $k=10$     |  0.90   |  1.00   |  1.00   |     1.00      |   1.00   |   1.00   |   1.00   |
> |    $k=50$     |  0.94   |  0.96   |  0.96   |     1.00      |   0.98   |   1.00   |   0.98   |
> |    $k=100$    |  0.88   |  0.90   |  0.93   |     1.00      |   0.95   |   0.94   |   0.94   |
>
> ### **4. Response to the question of hyperparameters.**
> Thank you for your question regarding the hyperparameters $\alpha$ and $K$. In our experiments, we conducted a grid search to identify the optimal values. We find that tuning both $\alpha$ and $K$ is necessary, providing a more fine-grained control over the intervention. The results show that CAI achieves the best hallucination mitigation performance when $\alpha=1.5$ and $K=100$.
>
> To further address your concern, we now additionally provide the full grid-search results of **LLaVA-1.5-7B** on the POPE-Adversarial benchmark. These results give a more clear and continuous view of how $\alpha$ and $K$ jointly affect CAI’s performance.
>
> | Accuracy | $\alpha=0$ | $\alpha=0.5$ | $\alpha=1.0$ | $\alpha=1.25$ | $\alpha=1.5$ | $\alpha=1.75$ | $\alpha=2.0$ |
> | -------- | :--------: | :----------: | :----------: | :-----------: | :----------: | :-----------: | :----------: |
> | $K=0$    |   78.96    |    78.96     |    78.96     |     78.96     |    78.96     |     78.96     |    78.96     |
> | $K=50$   |   78.96    |    79.31     |    79.86     |     80.18     |    80.50     |     80.32     |    80.40     |
> | $K=75$   |   78.96    |    79.82     |    80.13     |     80.44     |    80.77     |     80.59     |    80.31     |
> | $K=100$  |   78.96    |    81.07     |    82.50     |     83.47     |  **84.27**   |     84.14     |    84.00     |
> | $K=125$  |   78.96    |    80.79     |    82.16     |     83.28     |    84.10     |     83.82     |    83.51     |
> | $K=150$  |   78.96    |    80.24     |    81.47     |     82.53     |    83.20     |     82.97     |    82.68     |
> | $K=200$  |   78.96    |    79.91     |    81.18     |     82.12     |    83.00     |     82.76     |    82.43     |
>
> ### **5. Response to nitpicks.**
> Thank you for your careful and detailed inspection. We will further clarify these details in the revision to avoid ambiguity:
> - **Figure 5:**  As stated in line 269, Figure 5 reports the SVM classifier accuracies trained on 1,000 VQAs (from the LLaVA-1.5-7B pretraining dataset). The two curves correspond to **LLaVA-1.5-7B** and **Qwen-VL-Chat**, respectively.
> - **Figure 6:**  $K = 100$ on the left and $\alpha = 1.5$ on the right (with the other hyperparameter fixed to its optimal value, as indicated by our grid search). As labeled in the legend, the y-axis values represent the accuracy and F1-score of LLaVA-1.5-7B and Qwen-VL-Chat on the POPE-Adversarial benchmark.
> - Regarding the note in line 197: As you pointed out, we will correct this typo.
>
> ---
>
> ### _**We sincerely thank for the reviewer‘s careful reading and insightful feedback. We hope these clarifications could address your concerns.**_

---

### Official Review · Reviewer_M7q8 · 2025-11-03

**Soundness:** 3
**Presentation:** 3
**Contribution:** 3
**Rating:** 6
**Confidence:** 3

**Summary:**

This paper introduce CAI, a training-free method to mitigate a training-free method to mitigate object hallucination in Large Vision-Language Models. The authors first identify that caption queries uniquely amplify visual attention in particular attention heads compared to non-caption queries, endowing the model with fine-grained visual perception capability. Leveraging this phenomenon, CAI employs a three-step inference-time intervention: (1) probing these caption-sensitive heads via SVM classifiers, (2) computing perception-refined vectors as shift directions between caption and non-caption responses, and (3) steering attention outputs using intensity parameters.  Evaluations across five benchmarks show CAI reduces hallucination by 6.03% on average, achieving SOTA performance while preserving foundational capabilities and minimizing inference latency. This work contributes novel insights into attention optimization, a practical intervention framework, and strong generalizability across models and tasks.

**Strengths:**

- This paper is the first to explicitly reveal that caption queries uniquely enhance visual attention in specific LVLM attention heads. This discovery is substantiated by quantitative evidence and layer-wise analysis, demonstrating that the attention amplification correlates with reduced object hallucination and provides valuable insights into fine-grained visual perception mechanisms.
- The proposed method demonstrates state-of-the-art performance across different benchmarks while maintaining strong generalizability across LVLM architectures. It achieves superior hallucination mitigation with significantly lower inference latency, ensuring practical efficiency without performance compromises.

**Weaknesses:**

- The study does not adequately address how variations in non-caption queries affect the proposed method. The probing methodology relies on a limited set of non-caption queries, potentially leading to inaccurate perception-refined vectors when handling diverse real-world queries. This limitation may compromise the method's robustness in practical applications.

**Questions:**

- This paper uses SVM for probing caption-sensitive attention heads. Have the authors considered or experimented with alternative classification methods ?
- The analysis of hyperparameters is currently conducted only on the POPE benchmark. I believe additional validation across diverse benchmarks is necessary, as different task types may exhibit distinct parameter sensitivities.

---

> ### Author Response · Authors · 2025-11-19
>
> We thank the reviewer for the insightful feedback. Next, we will explain your concerns point by point:
> ### **1. Response to the concern on practical applications.**
> Thank you for raising concerns regarding practical applications, and we have provided careful design and experimental validation for the generality of CAI:
>
> - **From the perspective of classifier training data:** As described in Section 4.1 and Appendix B.2.2, we use 1,000 **_task-diverse_** VQA questions sampled from the LLaVA-1.5-7B pretraining dataset as our non-caption queries. These queries cover the vast majority of scenarios likely to be encountered in real-world applications.
> - **From the perspective of CAI’s core design:** The central mechanism of CAI (Equation 8) computes the **average attention difference** across these queries to obtain the _perception-refined vector_. This design ensures that the vector represents a **general direction** for fine-grained visual perception enhancement, rather than overfitting to any single type of non-caption query.
> - **From the perspective of empirical validation:** Although derived exclusively from LLaVA’s pretraining data, the CAI vector yields consistent gains across diverse LVLM architectures and hallucination benchmarks, and further extends effectively to domain-specific tasks (e.g., VQA-RAD, MMBench OCR; App. F), indicating strong robustness in practical applications.
>
> ### **2. Discussion of the classifier types.**
> Thank you for raising this question. Inspired by prior work [1,2,3] showing that SVM effectively performs binary classification on high-dimensional internal model vectors, we adopt SVM as the classifier in our CAI framework.
>
> To further address your concern, we implement **Logistic Regression (LR)** as an alternative classifier. The experimental results are shown in the table below. CAI with LR achieves performance nearly identical to CAI with SVM, as **95% of the top-100 attention heads selected by both classifiers are the same**. CAI with SVM exhibits a slight performance advantage, which aligns with findings reported in the related work [4] and further confirms SVM’s superior capability in classifying high-dimensional vectors.
>
> |              | POPE ran. | POPE pop. | POPE adver. | CHAIRi$\downarrow$ | CHAIRs$\downarrow$ |          MME(count\|exist.\|color\|posi.)           |
> | :----------- | :-------: | :-------: | :---------: | :----------------: | :----------------: | :--------------------------------------------------: |
> | LLaVA-1.5-7b |   83.29   |   81.88   |    78.96    |        15.9        |        52.8        |         124.67 \| 175.67 \| 151.00 \| 114.00         |
> | + CAI w/ SVM | **89.87** | **88.32** |  **84.27**  |      **11.5**      |      **34.6**      | **141.67** \| **190.00** \| **170.00** \| **140.00** |
> | + CAI w/ LR  |   89.40   |   88.13   |    83.87    |        11.7        |        34.9        |     138.33 \| **190.00** \| **170.00** \| 135.00     |
>
> [1] Inference-Time Intervention: Eliciting Truthful Answers from a Language Model
>
> [2] Probing the Geometry of Truth: Consistency and Generalization of Truth Directions in LLMs Across Logical Transformations and Question Answering Tasks
>
> [3] PIP: Detecting Adversarial Examples in Large Vision-Language Models via Attention Patterns of Irrelevant Probe Questions
>
> [4] ThoughtProbe: Classifier-Guided LLM Thought Space Exploration via Probing Representations

---

> ### Author Response · Authors · 2025-11-19
>
> ### **3. Additional analysis of hyperparameters.**
> Thank you for raising concerns regarding the hyperparameter analysis. We performed fine-grained parameter search only on the POPE dataset and observed that the optimal parameters identified on this dataset **can generalize well to other discriminative and generative tasks** (e.g., MME, CHAIR, MMHal-Bench). This indicates that the fixed optimal hyperparameters can be effectively applied in real-world scenarios, demonstrating **CAI’s ease of deployment and strong generalization capability**.
>
> Furthermore, we agree with your observation that _“different task types may exhibit distinct parameter sensitivities.”_ Indeed, CAI can achieve slightly better performance with task-specific hyperparameters in some generative tasks. As shown in the table, we conducted hyperparameter analysis on the CHAIR benchmark. The optimal parameters are found to be ($\alpha = 1.25$, $K = 125$ and performance = *34.3*); nevertheless, the performance difference compared to the POPE-optimal parameters ($\alpha = 1.5$, $K = 100$ and performance = *34.6*) is minimal.
>
> | CHAIRs$\downarrow$ | $\alpha = 0$ | $\alpha = 1.0$ | $\alpha = 1.25$ | $\alpha = 1.5$ | $\alpha = 1.75$ | $\alpha = 2.0$ |
> | ------------------ | :----------: | :------------: | :-------------: | :------------: | :-------------: | :------------: |
> | $K = 0$            |     52.8     |      52.8      |      52.8       |      52.8      |      52.8       |      52.8      |
> | $K = 50$           |     52.8     |      44.3      |      43.1       |      43.5      |      44.0       |      44.8      |
> | $K = 75$           |     52.8     |      39.6      |      37.5       |      37.6      |      38.6       |      39.4      |
> | $K = 100$          |     52.8     |      35.1      |      34.4       |     *34.6*     |      35.2       |      35.9      |
> | $K = 125$          |     52.8     |      34.9      |    **34.3**     |      34.5      |      35.0       |      35.7      |
> | $K = 150$          |     52.8     |      35.3      |      34.7       |      35.1      |      35.8       |      36.5      |
> | $K = 200$          |     52.8     |      36.0      |      34.4       |      36.1      |      36.7       |      37.3      |
>
> ---
>
> ### _**We sincerely thank for the reviewer‘s careful reading and insightful feedback. We hope these clarifications could address your concerns.**_

---

### Author Response · Authors · 2025-11-25
**Summary of Revisions**

Dear PCs, SACs, ACs, and Reviewers,

We appreciate the reviewer’s constructive and insightful feedback. To address the valuable comments raised by the reviewers, we have made the following adjustments to the paper. (We have highlighted the technical revisions in blue.)

---
**Main revisions:**
 1. **[Reviewer mtt9: Quick Study of Confusion Matrix] (Appendix L)** We cite the related work provided by the reviewer and compute the confusion matrices of LLaVA-1.5-7B on the POPE _popular_ and _random_ subsets, which shows CAI substantially mitigates the “yes-bias”.
 2. **[Reviewer mtt9: Discussion of CAI on the Fly] (Appendix J)** We appreciate the reviewer’s interesting suggestion of applying CAI using an _on-the-fly_ strategy, rather than relying on _precomputed_ shift vectors as in our main approach. However, we find the _on-the-fly_ strategy not only exhibits slightly lower perfomance than  _precomputed_ strategy, but also increases inference latency.
 3. **[Reviewer mtt9: Additional Experimental Results of Advanced Models, Methods and Benchmarks] (Appendix C, D&G)**

    (1) We refine Appendix C and add the full results of more advanced models (LLaVA-NeXT, Qwen2.5-VL-7B), beyond the Qwen2-VL-7B and InternVL2-8B models already included in the paper.

    (2) We refine Appendix D and add comparisons between CAI and two RL-based methods (HADPO and HALVA), beyond the 13 training-free methods originally reported.

    (3) We create Appendix G and add experiments on more benchmarks (HallusionBench and GAVIE), beyond the five benchmarks already presented in the paper.
 4. **[Reviewer bUqS: Discussion on the CAI Intervention Layers] (Appendix K)** We appreciate the reviewer’s excellent insight. We control the index of layers on which interventions are applied and find that important attention heads must be activated or perturbed across layers to fully reinforce the visual information flow, leading to better performance.
 5. **[Reviewer M7q8: Impacts of the Classifier Types] (Appendix H.1)** We implement Logistic Regression as an alternative classifier and find that the SVM classifier demonstrates superior performance.
 6. **[Reviewer bzdp: Impacts of the Classifier Training Datas] (Appendix H.2)** We randomly select distinct samples from the whole LLaVA-1.5-7B pre-training dataset (77K) and retrain the classifiers. Experiments show that the top-k attention heads identified from training on different amounts of data exhibit strong consistency.
 7. **[Reviewer bzdp, M7q8: Fine-grained Analysis of Hyperparameters] (Appendix I)**
We provide the full grid-search results of LLaVA-1.5-7B on the POPE adversarial subset to observe the joint effect of hyperparameters. Besides, we additionally conduct hyperparameter analysis on the CHAIR benchmark and find that the optimal parameters are similar to those found on POPE, and their performance is comparable.
---
**Other minor revisions:**
- Revised Fig. 5 (line 394) and Fig. 6 (line 422) to improve clarity.
- Revised typo of $b'$ in line 197.

Again, thank you for the time and thoughtful reviews you devoted to this paper. Your efforts have made this paper stronger.

Authors

---

### Author Response · Authors · 2025-11-30
**Summary of Rebuttal Phase**

**Dear PCs, SACs, ACs, and Reviewers,**

We sincerely thank you for your time and dedication to the review process. In light of the recent Leak Incident and the resulting rollback of scores, we would like to provide a factual summary of our rebuttal progress and the outcomes **(the ratings universally reached 6, 6, 6, 6 two days before the incident)**. We affirm that the following details are accurate and adhere to academic integrity, intended to assist you in making a fair decision.

---
- **Comprehensive Responses:** We provided detailed, point-by-point responses to every question raised by the reviewers, supported by clear logic and comprehensive additional experiments. During the rebuttal, **Reviewer mtt9 raised the Rating from 2 to 6 and Reviewer bUqS made a positive response**.
- **Meticulous Revisions:** We have diligently revised the paper, incorporating reviewers' most feedback and aligning the changes with our rebuttal responses. More details are available in **Summary of Revisions**.
- **Rating Increasing (Reviewer mtt9):** **Reviewer mtt9** updated the evaluation on **\[Nov 25, 2025, 07:08 AOE\]** (well before the Leak Incident occurred), and explicitly acknowledged the quality of our rebuttal and **raised the Rating  from 2 to 6** (aligning with other reviewers). You can verify this Rating Increasing via the 'Revisions' history button on this Official Review ([Openreview Link](https://openreview.net/revisions?id=CM6x87OhxR)). This improvement reflects that our response successfully clarified the reviewer's misunderstandings regarding the abstract and experimental setup, and addressed all concerns with sufficient additional results.
- **Positive Response (Reviewer bUqS):** **Reviewer bUqS** posted a comment on **\[Nov 27, 2025, 02:13 AOE\]** (shortly before the Leak Incident occurred), stating: _"Most of my concerns are addressed, and I maintain my positive rating."_ This confirms their continued endorsement of our work's quality and the effectiveness of our rebuttal.
- **Initial Positive Reviews:** **Reviewers M7q8 and bzdp** initially provided **positive ratings (6)**, acknowledging the value of our work. Although they did not participate in further discussion during this period, we believe our comprehensive rebuttals and additional experiments can mostly resolve their stated concerns.
---
We have invested significant time and effort during the rebuttal phase to address all reviewers' concerns and improve the manuscript. We sincerely hope the ACs, SACs, and PCs could make a fair judgment based on these factual records, minimizing the impact of this incident.

Thank you for your time and consideration.

**Best regards,**

**Authors**

---

### Meta-Review · Area_Chair_NTWf · 2025-12-22

**Summary:**

This paper introduces Caption-Sensitive Attention Intervention (CAI), a training-free framework designed to mitigate object hallucination in Large Vision-Language Models (LVLMs). The core premise relies on the observation that LVLMs exhibit stronger visual attention activation when processing caption-based queries compared to non-caption queries. To exploit this, the authors propose a pipeline that uses offline probing (via SVMs) to identify "caption-sensitive" attention heads and subsequently applies an inference-time shift vector to intervene in the attention mechanism, thereby enhancing visual grounding.

Reviewers initially recognized the interesting empirical observation regarding attention patterns and the method's effectiveness on standard hallucination benchmarks like POPE and CHAIR. During the rebuttal, the authors made commendable efforts to address concerns by including stronger baselines (e.g., LLaVA-NeXT) and providing confusion matrix analyses to clarify "Yes-bias" issues. However, significant weaknesses remain regarding the method's generalizability and safety. Specifically, reviewers pointed out that the intervention negatively impacts text-centric capabilities (e.g., machine translation), and the proposed solution (adding a secondary textual probe) introduces excessive complexity. Furthermore, the reliance on sensitive hyperparameters ($\alpha, K$) and the heuristic nature of the intervention limit the contribution's robustness. Consequently, the Area Chair (AC) recommends rejection.

**Reviewer Concerns:**

- Reviewer bzdp highlighted a critical flaw: the method significantly degrades performance on text-only tasks, such as translation. While the authors proposed a fix during rebuttal—adding a parallel "Textual Probe" to filter out text-sensitive heads—this creates a complex trade-off. The AC agrees that this "patching" approach suggests the visual and textual representations are not cleanly disentangled, and the method risks overfitting to hallucination metrics at the expense of the model's foundational linguistic competence.
- Reviewers bzdp and M7q8 expressed concern over the tuning of the intervention strength ($\alpha$) and the number of heads ($K$). The authors' additional analysis showed that optimal parameters drift between discriminative tasks (like POPE) and generative tasks (like CHAIR). This contradicts the claim of a "plug-and-play" solution, as deployment in real-world scenarios would arguably require task-specific grid searches, reducing practicality.
- Reviewer bUqS raised a valid theoretical concern regarding whether the manually shifted hidden states remain within the model's valid latent manifold. The intervention is linear and heuristic; without formal constraints, there is a lingering risk that such perturbations could introduce undefined behaviors or artifacts in long-tail scenarios, which was not theoretically addressed.

**Reviewer Scores:**

- The method is viewed as an engineering heuristic derived from a specific observation (captioning attention) rather than a principled method for correcting multimodal alignment. The necessity to train classifiers (offline) and compute shift vectors makes it less elegant than dynamic decoding strategies.
- Reviewers remain unconvinced that the +6% gain on specific hallucination benchmarks justifies the risk of breaking other model capabilities. The "fix" for translation issues introduced in the rebuttal adds architectural complexity (dual probes), which reviewers perceive as an ad-hoc adjustment rather than a robust methodological contribution.

---

### Decision · Program_Chairs · 2026-01-26

Reject